# A direct comparison of theory-driven and machine learning prediction of suicide: A meta-analysis

**Katherine M. Schafer**[1]*, **Grace Kennedy**[1,2], **Austin Gallyer**[1], **Philip Resnik**[3]

1 Florida State University, Department of Psychology, Tallahassee, Florida, United States of America,
2 Walter Reed National Military Medical Center, Bethesda, Maryland, United States of America,
3 Department of Linguistics and Institute for Advanced Computer Studies, University of Maryland, College Park, Maryland, United States of America

* kmm16n@fsu.edu

**Data Availability Statement:** All relevant data are within the paper and its Supporting Information files.

## Abstract

Theoretically-driven models of suicide have long guided suicidology; however, an approach employing machine learning models has recently emerged in the field. Some have suggested that machine learning models yield improved prediction as compared to theoretical approaches, but to date, this has not been investigated in a systematic manner. The present work directly compares widely researched theories of suicide (*i.e.*, BioSocial, Biological, Ideation-to-Action, and Hopelessness Theories) to machine learning models, comparing the accuracy between the two differing approaches. We conducted literature searches using PubMed, PsycINFO, and Google Scholar, gathering effect sizes from theoretically-relevant constructs and machine learning models. Eligible studies were longitudinal research articles that predicted suicide ideation, attempts, or death published prior to May 1, 2020. 124 studies met inclusion criteria, corresponding to 330 effect sizes. Theoretically-driven models demonstrated suboptimal prediction of ideation (wOR = 2.87; 95% CI, 2.65–3.09; $k = 87$), attempts (wOR = 1.43; 95% CI, 1.34–1.51; $k = 98$), and death (wOR = 1.08; 95% CI, 1.01–1.15; $k = 78$). Generally, Ideation-to-Action (*w*OR = 2.41, 95% CI = 2.21–2.64, $k = 60$) outperformed Hopelessness (*w*OR = 1.83, 95% CI 1.71–1.96, $k = 98$), Biological (*w*OR = 1.04; 95% CI .97–1.11, $k = 100$), and BioSocial (*w*OR = 1.32, 95% CI 1.11–1.58, $k = 6$) theories. Machine learning provided superior prediction of ideation (wOR = 13.84; 95% CI, 11.95–16.03; $k = 33$), attempts (wOR = 99.01; 95% CI, 68.10–142.54; $k = 27$), and death (wOR = 17.29; 95% CI, 12.85–23.27; $k = 7$). Findings from our study indicated that across all theoretically-driven models, prediction of suicide-related outcomes was suboptimal. Notably, among theories of suicide, theories within the Ideation-to-Action framework provided the most accurate prediction of suicide-related outcomes. When compared to theoretically-driven models, machine learning models provided superior prediction of suicide ideation, attempts, and death.

**Funding:** This work was supported by a grant from the National Institute of Mental Health (T32MH093311).

**Competing interests:** The authors have declared that no competing interests exist.

## Introduction

Suicide is a leading cause of death worldwide, a global epidemic that kills nearly 800,000 people every year [1]. Despite widespread investigation into suicidal thoughts and behaviors (STBs), suicide rates have failed to appreciably decline in the United States [2]. Notably this is not the trend worldwide, as many nations across the world have witnessed significant decreases in rates of suicide [1]). Fully understanding the causal mechanisms of suicidal ideation, attempts, and death has been a long-standing goal for suicide research and practice, and the field has traditionally taken the approach of proposing theories to understand these causal mechanisms. Theories of suicide generally suggest that one or a small set of biopsychosocial factors combine together to lead to suicidal ideation, attempts, and death (*e.g.*, Biological, [3]; Hopelessness Theory, [4]; Ideation-to-Action, [5–7] BioSocial Theory of Suicide, [8]). However, in recent years, a new approach to suicide research has arisen in the field, aimed at accurate prediction of suicidal ideation, attempts, and death, and rooted in machine learning. Nonetheless, there has yet to be a systematic empirical investigation determining if theoretically-driven models or machine learning models are preferable in predicting suicidal ideation, attempts, and death. The present project addresses this crucial gap.

### Theories of suicide

Throughout decades of suicide research, many theories have been proposed in the effort to understand the causal mechanisms of suicidal ideation, attempts, and death. These theories include the Biological Theory of Suicide [3], the Hopelessness Theory of Suicide [4], the Ideation-to-Action perspective of suicide [5–7], and the BioSocial Theory [8]. The (1) Biological Theory of Suicide [3] represents a body of literature that generally posits a diathesis-stress model for the development of STBs. Researchers have proposed that biomarkers for suicide have been linked to altered stress responses and to abnormalities in the serotonergic system. It is the hope that these biological causes of STBs can be identified and eventually altered to reduce STBs. Likewise, (2) the Hopelessness Theory of Suicide [4] proposes that depression is closely linked with STBs, and that hopelessness acts as the avenue leading people suffering from depression to engage in STBs. The Interpersonal [5], Three-Step [6], and the Motivational-Volitional [7] Theories of Suicide all cluster into (3) the Ideation-to-Action perspective of suicide. This approach to suicidology seeks to identify the causes of suicidal ideation as well as understand what factors lead to the shift from ideation to action (*i.e.*, suicide attempts and death). These theories propose that, among many constructs, thwarted belongingness, perceived burdensomeness, acquired capability, loneliness, defeat, and entrapment are the causes of STBs. Finally, (4) the BioSocial Theory has its roots in the treatment of patients with Borderline Personality Disorder. This approach to suicide generally proposes that emotion dysregulation, caused by a genetic vulnerability coupled with an invalidating environment, leads to STBs. Notably, in this approach, STBs represent a method of regulating out-of-control emotions.

The Biological, Hopelessness, Ideation-to-Action, and BioSocial theories of suicide have each garnered support in the literature, with many of these theories using the same, or similar, constructs to predict suicide-related outcomes. (The Biological Theory represents a departure from this trend, in that it proposes that constructs related to hormones, neurotransmitters, and specific brain regions may be at play for suicide-related outcomes). Many studies have indicated that constructs related to Hopelessness theory (*i.e.*, depressive symptoms and hopelessness [*e.g.*, [9,10]]), Biological risk factors (*e.g.*, saturation of serotonin and dopamine [11–13], Ideation-to-Action (*e.g.*, entrapment, thwarted belongingness, perceived burdensomeness, loneliness, acquired capability, and lack of connection [*e.g.*, [14–16]]), and BioSocial theory (*e.*

*g.*, emotion dysregulation, dysregulation of negative affect [*e.g.*, [17,18]) are cross-sectionally and longitudinally associated with STBs. Yet, although these findings generally support correlations predicted by theories of suicide, they have not been meta-analyzed in a way that investigates the overall prediction of suicide-related outcomes as proposed by these theories. That is, the meta-analytic accuracy of longitudinal investigations of theoretically-relevant constructs grouped together as theories have not been compared. It is important that meta-analyses include longitudinal investigations, as theories were proposed in efforts to understand causal mechanisms that lead to the development of suicide-related outcomes. As cross-sectional studies do not estimate predictive accuracy, they contribute relatively little understanding of the causal role or the predictive accuracy of theoretically-relevant risk factors for suicidal ideation, attempts, and death.

## Machine learning in suicidology

Machine learning models of STBs have emerged in the field of suicidology (see [19] for a discussion of this emergence) as an alternative approach to the study of suicide. Whereas theories of suicide seek to understand causal mechanisms of suicidal ideation, attempts, and death, machine learning models—attempting a related goal—instead aim to accurately predict or classify these suicide-related outcomes. In other words, where theories of suicide seek to understand causal mechanisms, machine learning models seek to optimize prediction. (See also [20] for an argument that focusing on prediction as a goal is, in fact, a good idea even when the goal is to advance explanation and understanding.) Machine learning approaches generally use large datasets–*i.e.* large number of variables with large number of participants– and complex mathematical models to predict STBs. Generally speaking, such approaches are supervised, meaning that they use data with correct (or presumed-correct) labeling for a suicide-related independent variable to create a model based on observed independent variables (or complex combinations of variables, often also involving latent variables) that will lead to accurate classification for that variable. Creating the model generally involves optimizing some measure of goodness of the model, such as the proportion of "hits" (true-positive suicide-related outcomes) relative to false alarms or missed cases. Crucially, optimizing the model makes use of one set of data, usually referred to as "training data" and then these algorithms are tested on unseen or new data, referred to as "test data".

Machine learning models can be used cross-sectionally to classify (*e.g.*, identify) participants who endorse suicide-related outcomes at the time of data collection. For example, it is possible for researchers to collect data using multiple psychological self-report measures, physiological measures, and demographic features and construct a machine learning algorithm to classify suicide-related outcomes, all from data collected at the same time point. Alternatively, machine learning models can be used to longitudinally predict who will develop suicide-related outcomes at a point later in time. For example, a machine learning algorithm can be trained to use decades of health care or medical records to predict who will develop suicidal ideation at a later point, or to use several months' worth of social media posts to predict who will actually make a suicide attempt. Thus, machine learning models can be used to classify (*e.g.*, cross-sectionally) or predict (*i.e.*, longitudinally) cases of suicide-related outcomes. Notably, theoretically-driven models do not provide learning and testing on two disparate groups of data [20].

While machine learning methods have existed for a long time, they have recently received greater attention in the study of suicide and several recent studies have used machine learning models to predict STBs from optimized combinations of large sets of factors (Walsh et al., 2018). In general, studies report that machine learning models produce highly accurate

prediction of STBs (odds ratios greater than 1000), that are vastly more accurate than theoretical approaches. A machine learning approach to suicide prediction is therefore promising, and has been the focus of considerable meta-analytic design (see [21–23]), yet previous work has not investigated or compared the predictive ability of machine learning approach to long-standing traditional methods. Thus, the question of whether the accurate novel prediction of machine learning is any improvement upon traditional theories of suicide remains to be rigorously established.

## The present study

The present work represents a direct comparison of two approaches to the study of suicidology: theoretically-driven models versus machine learning models. Predictive ability of theoretically-driven models of STBs has not been comprehensively investigated, particularly in regards to Biological Theory [3], Hopelessness Theory [4], Ideation-to-Action Theories [5–7], or the BioSocial Theory [8]. Meta-analyses of traditional STB risk factors have been highly general and/or examined disorder-specific predictors [24–29]. That is, these general meta-analyses of STB risk factors have been too broad to compare the relative accuracy of theoretically-driven models. Likewise, although previous meta-analytic work has assessed machine learning prediction of STBs [21–23], studies have failed to compare this prediction to the theoretically-driven approach. Thus, our work builds on previous meta-analytic work on machine learning approaches to STB prediction by comparing the two approaches. Importantly, the present work examines the predictive ability of the two opposing approaches using odds ratios as a common metric.

Our study was conducted with two aims in mind. The first aim of our study is to summarize and compare the predictive accuracy of various theories of STBs. Specifically, we compare accuracy of prediction of suicidal ideation, attempts, and death within and between four popular theories of suicide (*i.e.* Biological, Hopelessness, Ideation-to-Action, and BioSocial Theories). The second aim of the present study is to compare the ability of machine learning and theoretically-driven models to predict STBs.

## Methods

### Sources and searches

The present meta-analysis was completed in accordance with PRISMA guidelines [30]. The institutional review board of the Florida State University approved this study. Literature searches using PubMed, PsycINFO, and Google Scholar were conducted to include studies published between January 1, 1900 to May 1, 2020. Theoretically-driven models were collected by entering search strings that were combinations of variants of longitudinal (for example longitudinal, predicts, prediction, prospective), suicide (for example suicide, suicidal ideation, suicide attempt, and suicide death), and constructs related to four popular theoretical perspectives of suicide (1) a Biological perspective [3], (2) Hopelessness theory [4], (3) Ideation-to-Action framework [5–7], and (4) BioSocial [8]. See Supporting Information section (S1–S3 Tables) for an exhaustive list of search terms. Machine learning models were collected by searching for variants of suicide (for example suicide, suicidal ideation, suicide attempt, and suicide death) with variants of machine learning (machine learning, artificial intelligence, data mining, computer applications, data processing). We also searched reference sections of all papers identified through our database searches. Finally, we searched for unpublished literature by searching for conference postings and emailing corresponding authors for unpublished data. Of an additional 26 machine learning manuscripts that appeared to meet eligibility

criteria but were missing sufficient data, when contacted, only 5 of the corresponding authors replied with sufficient data.

## Study selection, inclusion criteria

Inclusion criteria for theoretically-driven models required that papers be in English and include at least one longitudinal analysis predicting suicidal ideation, attempts, or death using any variable related to theoretical constructs. Given the paucity of literature related to the machine learning perspective, inclusion criteria for machine learning papers included cross-sectional, longitudinal, and experimental study design.

Papers were excluded if (a) no analyses examined the discrete effects on suicidal ideation, suicide attempts, or deaths by suicide, or (b) analyses were conducted within a primary treatment study. The initial search terms yielded a total of 596 publications. Based on inclusion criteria, 124 papers were retained. (See Fig 1 for PRISMA flowchart and Table 1 for list of included studies by paradigm and outcome).

## Data extraction and coding

Any statistical test that used a variable related to theoretical constructs to predict suicidal ideation, suicide attempt, or death by suicide was retained for analysis, with a total of 330 unique effect sizes. Data extracted from each study included: authors, publication year, number of participants with STB outcome, predictor variable, outcome variable, and any relevant statistics from each prediction case. These effect sizes were initially independently extracted by the first author and a bachelor's level graduate student in psychology. Discrepancies were resolved through discussion until consensus. The resultant two-way intraclass correlation (ICC) was excellent (ICC = .978).

**Theoretically-driven versus machine learning basis.** The primary analysis was the comparison of theoretically-driven models and machine learning accuracy. Effect sizes were coded: (a) theoretically-driven and (b) machine learning. Authors compared the effect sizes in odds ratios of theoretically-driven versus machine learning framework, determining if a particular framework was significantly more accurate in STB prediction.

**Theoretical basis.** Within the traditional paradigm, four theoretical bases were studied. Effect sizes were coded as: (a) Biological framework, (b) Hopelessness theory, (c) Ideation-to-Action framework, and (d) BioSocial Theory. Machine learning models do not have a formalized theory; accordingly, machine learning models were coded as (e) machine learning models. Using this coding scheme, we compared the effect sizes between the theoretical bases.

*Study design.* Inclusion criteria for traditional effect sizes required longitudinal prediction. Machine learning had fewer effect sizes in the extant literature, necessitating inclusion of cross-section and longitudinal designs. Using the study design code, authors compared accuracy of machine learning models based on cross-sectional or longitudinal design. The code related to study design was as follows: (a) correlational and (b) longitudinal.

**Suicide outcome.** The STB outcome variable for each effect size was coded as follows: (a) suicidal ideation, (b) suicide attempt, and (c) suicide death. This code was collected to test if theoretically-driven models or machine learning models change predictive power based on the outcome type. Notably, if studies combined outcomes, those studies were excluded from analyses, and thus were not coded.

**Year.** The year of publication was recorded for each effect size. Authors conducted analyses to test if the effect sizes of conceptualizations have changed significantly over time.

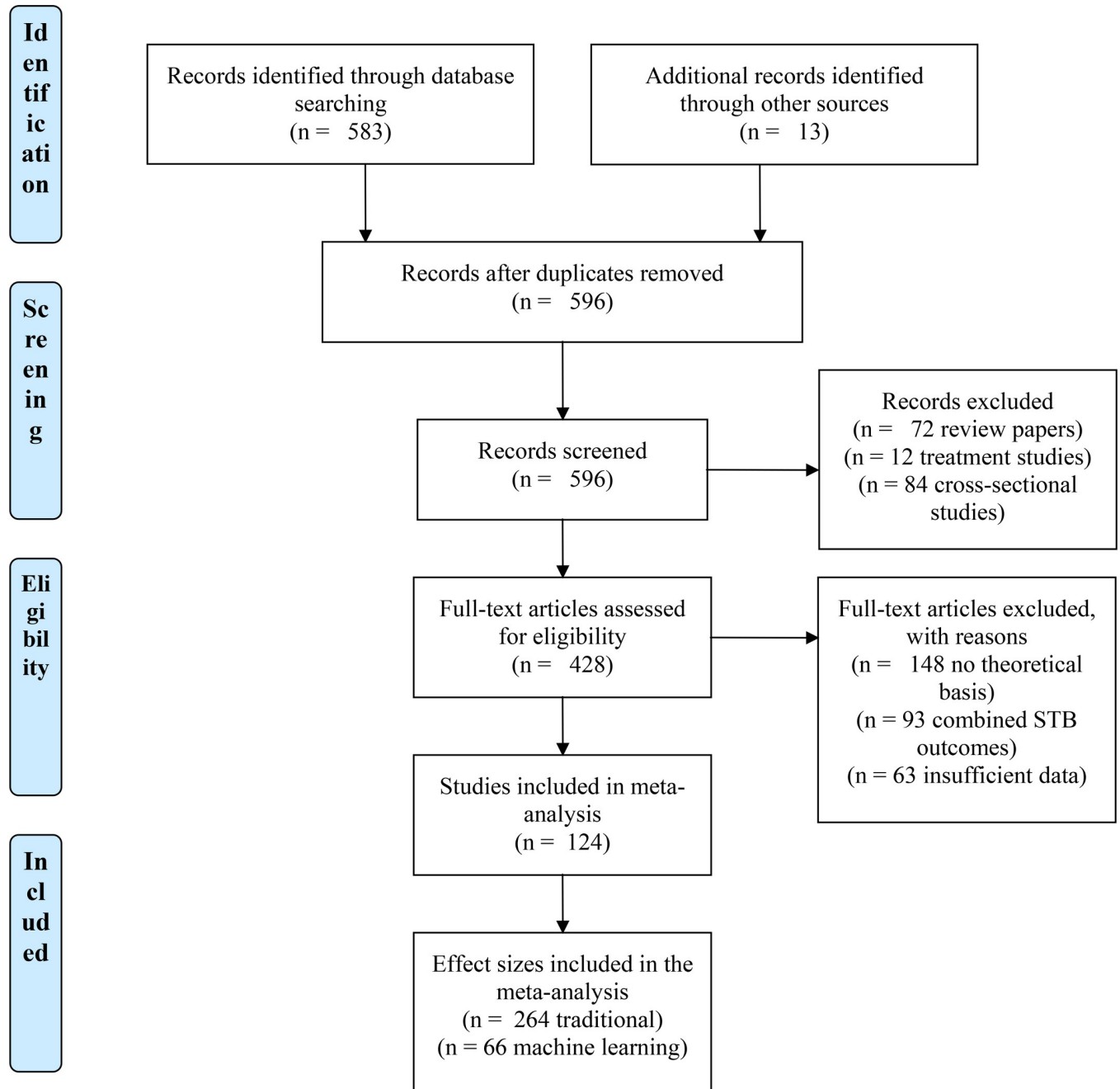

**Fig 1. PRISMA diagram for studies included in the present analysis.**

## Statistical analyses

Meta-analyses were performed using Comprehensive Meta-Analysis (CMA), Version 3. Unadjusted effects were used as this provided the purest effect estimate, as unadjusted effect sizes do not control for extraneous variables. Unadjusted effect sizes allow for the most accurate between-study comparisons. CMA converted all effects sizes to a common metric: odds ratio (OR). OR provides a numerical value connecting exposure to a variable and an outcome. In the present study this can be conceptualized as experiencing thwarted belongingness,

**Table 1. List of studies by paradigm and outcome.**

| Study Name | Paradigm | Outcomes | Quality |
|---|---|---|---|
| | *Biological* | | |
| Abbar et al (2001) | | Attempt | 1 |
| Asberg et al (1976) | | Attempt, Death | 1 |
| Bellivier et al (2000) | | Attempt | 1 |
| Black et al. (2002) | | Attempt, Death | 1 |
| Chatzittofis et al (2013) | | Death | 1 |
| Chong et al (2000) | | Attempt | 1 |
| Clark (1981) | | Attempt | 1 |
| Consoloni et al. (2018) | | Ideation, Attempt | 1 |
| Coryell and Schlesser (1981) | | Death | 1 |
| Coryell and Schlesser (2001) | | Death | 1 |
| Coryell and Schlesser (2007) | | Death | 1 |
| Courtet et al (2001) | | Attempt | 1 |
| Crawford et al (2000) | | Death | 1 |
| Du et al. (1999) | | Death | 1 |
| Ellison and Morrison (2001) | | Death | 1 |
| Engstrom et al (1999) | | Attempt, Death | 1 |
| Fiedorwicz and Coryell (2007) | | Attempt | 1 |
| Giletta et al (2017) | | Ideation | 1 |
| Gorwood et al (2000) | | Attempt | 1 |
| Irribarren et al (1995) | | Death | 1 |
| Jokinen and Nordstrom (2008) | | Death | 1 |
| Jokinen et al (2009) | | Death | 1 |
| Jokinen et al (2012) | | Death | 1 |
| Jokinen et al (2007) | | Death | 1 |
| Keilp et al (2008) | | Attempt | 1 |
| Komaki et al (2008) | | Death | 1 |
| Lewis et al (2011) | | Death | 1 |
| Nordstrom et al (1994) | | Death | 1 |
| Roy et al (1986) | | Death | 1 |
| Roy et al (1989) | | Attempt | 1 |
| Roy et al (1992) | | Attempt | 1 |
| Samulsson et al (2006) | | Death | 1 |
| Sher et al (2006) | | Attempt | 1 |
| Targum et al (1983) | | Attempt | 1 |
| Traskman et al (1981) | | Death | 1 |
| Yerevanian et al (1983) | | Death | 1 |
| Yerevanian et al (2004) | | Attempt, Death | 1 |
| Zureik et al 1996 | | Death | 1 |
| | *Hopelessness* | | |
| Beautrais et al (2004) | | Attempt, Death | 2 |
| Beck (1985) | | Death | 2 |
| Beck (1989) | | Death | 2 |
| Bennardi et al (2019) | | Ideation | 2 |
| Blumental et al (1989) | | Death | 2 |
| Brent et al (2015) | | Attempt | 2 |
| Burke et al (2015) | | Attempt | 2 |

*(Continued)*

**Table 1.** (Continued)

| Study Name | Paradigm | Outcomes | Quality |
|---|---|---|---|
| Coryell and Schlesser (2001) | | Death | 2 |
| Dahlsgaard et al (1998) | | Death | 2 |
| Dieserud et al (2003) | | Attempt | 2 |
| Fawcett et al (1990) | | Attempt | 2 |
| Fulginiti et al 2018 | | Ideation | 2 |
| Handley et al (2016) | | Ideation | 2 |
| Holma et al (2014) | | Ideation, Attempt | 2 |
| Holma et al (2010) | | Attempt | 2 |
| Huth-bocks et al (2007) | | Attempt | 2 |
| Ialongo et al. (2004) | | Attempt | 2 |
| Keilp et al (2008) | | Attempt | 2 |
| Keller and Wolfersdorf (1993) | | Ideation | 2 |
| Kleiman and Beaver (2013) | | Ideation | 2 |
| Kleiman et al (2015) | | Ideation | 2 |
| Kuo et al (2004) | | Ideation, Attempt, Death | 2 |
| Lamis and Lester (2012) | | Ideation | 2 |
| Larzelere et al (1996) | | Attempt | 2 |
| Lecloux et al (2016) | | Attempt | 2 |
| Lecloux et al (2017) | | Attempt | 2 |
| Lewinsohn et al (1994) | | Attempt | 2 |
| Lewinsohn et al (2001) | | Attempt | 2 |
| Links et al. (2012) | | Ideation | 2 |
| Mars et al., 2019 | | Attempt | 2 |
| May et al (2012) | | Attempt | 2 |
| Miller et al 2016 | | Ideation | 2 |
| Miranda et al (2012) | | Attempt | 2 |
| Morrison and O'Connor (2008) | | Ideation | 2 |
| Mustanski and Liu (2013) | | Attempt | 2 |
| Nimeus et al (2000) | | Attempt | 2 |
| Nimeus et al 1997 | | Death | 2 |
| Nordentoft et al (2002) | | Attempt | 2 |
| O'Connor et al. (2013) | | Ideation | 2 |
| Pallaskorpi et al. (2016) | | Attempt | 2 |
| Panagioti et al (2015) | | Ideation, Attempt | 2 |
| Podlogar et al (2016) | | Ideation | 2 |
| Qiu et al (2017) | | Attempt | 2 |
| Quinones et al (2015) | | Ideation | 2 |
| Ribeiro et al (2012) | | Attempt | 2 |
| Ribeiro et al (2015) | | Death | 2 |
| Riihimaki et al (2013) | | Attempt | 2 |
| Robinson et al (2010) | | Death | 2 |
| Roeder and Cole (2018) | | Ideation | 2 |
| Samuelsson et al (2006) | | Death | 2 |
| Shi et al. (2018) | | Ideation | 2 |
| Sokero et al (2005) | | Attempt | 2 |
| Strange et al (2015) | | Ideation | 2 |
| Troister et al (2013) | | Ideation | 2 |

(Continued)

**Table 1.** (Continued)

| Study Name | Paradigm | Outcomes | Quality |
|---|---|---|---|
| Valtonen et al (2006) | | Attempt | 2 |
| Wang et al (2014) | | Attempt | 2 |
| Wilkinson et al (2011) | | Attempt | 2 |
| Young et al (1996) | | Attempt | 2 |
| | *Ideation-to-Action* | | |
| Batterham et al (2018) | | Ideation | 2 |
| Batterham et al (2017) | | Ideation | 2 |
| Bennardi et al (2017) | | Ideation | 2 |
| Bennardi et al (2019) | | Ideation | 2 |
| Brent et al (2015) | | Ideation | 2 |
| Chiu et al (2017) | | Ideation | 2 |
| Chu et al (2017) | | Ideation | 2 |
| Czyz et al (2015) | | Attempt | 2 |
| George et al., (2016) | | Ideation | 2 |
| Huang et al. (2017) | | Attempt | 2 |
| Kleiman and Beaver (2013) | | Ideation | 2 |
| Kleiman et al (2015) | | Ideation | 2 |
| Kuramoto-Crawford et al. (2017) | | Ideation | 2 |
| Lamis and Lester (2012) | | Ideation | 2 |
| Miller et al (2016) | | Ideation | 2 |
| Nielson et al (2015) | | Ideation | 2 |
| Panagioti et al (2015) | | Ideation | 2 |
| Podlogar et al (2016) | | Ideation | 2 |
| Puzia et al (2014) | | Ideation | 2 |
| Ribeiro et al (2015) | | Death | 2 |
| Roeder and Cole (2018) | | Ideation | 2 |
| Siguardson et al (2017) | | Ideation, Attempt | 2 |
| Strange et al (2015) | | Ideation | 2 |
| Van Orden et al (2015) | | Ideation | 2 |
| Zhang et al. (2018) | | Attempt | 2 |
| | *BioSocial* | | |
| Berglund (1984) | | Death | 2 |
| Links et al (2012) | | Ideation | 2 |
| Wedig et al (2012) | | Attempt | 2 |
| Wilcox et al (2010) | | Attempt | 2 |
| | *Machine Learning* | | |
| Cheng et al (2017) | | Ideation | 1 |
| Choi et al (2018) | | Death | 1 |
| Du et al (2018) | | Ideation | 1 |
| Fernandes et al (2018) | | Ideation, Attempt | 1 |
| Hill et al (2017) | | Ideation | 1 |
| Homan et al (2016) | | Ideation | 1 |
| Jung et al (2019) | | Ideation | 1 |
| Just et al (2017) | | Ideation | 1 |
| Kessler et al (2020) | | Ideation | 1 |
| Metzger et al (2016) | | Attempt | 1 |
| Naifeh et al (2019) | | Ideation | 1 |

*(Continued)*

**Table 1.** (Continued)

| Study Name | Paradigm | Outcomes | Quality |
|---|---|---|---|
| Oh et al (2020) | | Ideation | 1 |
| Passos et al (2016) | | Ideation | 1 |
| Ryu et al (2018) | | Ideation | 1 |
| Sanderson et al (2019) | | Death | 1 |
| Simon et al (2018) | | Attempt, Death | 1 |
| Walsh et al (2017) | | Attempt | 1 |
| Walsh et al (2018) | | Attempt | 1 |

hopelessness, emotion dysregulation, etc. and the associated STB outcome. When odds ratios (ORs) were not provided, CMA calculated them based on $2 \times 2$ contingency tables, correlations, independent group means and group sample size and t-statistics or p-value.

Between-study heterogeneity was quantified using I2-tests, which estimates the percentage of variation across studies due to heterogeneity rather than chance. Throughout the entirety of the data set, metrics of heterogeneity (I2 = 99.708 and Q = 108549.826), and as such, a random-effects model was used for all meta-analyses. Random-effects models assumed a distribution of effects across studies. The estimator used was restricted maximum likelihood. Thus, the combined effect estimated using a random-effects model represents the mean of the distribution of true effects rather than a single true effect (as in fixed-effects models). Using random-effects models, heterogeneity across studies is accounted for in the weighting and calculation of each effect. Small sample bias was examined throughout analyses using Egger's regression and inspection of funnel plots. Notably, we did not account for dependent effect sizes within the data. That is, for studies with multiple effect sizes corresponding to discrete suicide-related outcomes, we did not account for dependence between these effects.

For moderator analyses, meta-regression was employed using a random-effects model. Moderating effects were conducted related to paradigm (*i.e.*, traditional theories vs. machine learning), theory (*i.e.*, Biological vs. Ideation-to-Action vs. BioSocial vs. Hopelessness), STB outcome (*i.e.*, suicidal ideation vs. suicide attempt vs. suicide death), and year. Authors examined these potential moderation effects on overall prediction as well as on the effects of paradigm relevant constructs. Last, we examined small sample publication bias by visually inspecting funnel plots, and by using Duval and Tweedie's Trim and Fill and Egger's regression intercept.

## Results

Studies spanned 1976–2020 and included four traditional theories and machine learning studies. The most common theories were Biological theories (37.8%) followed by Ideation-to-Action (37.1%), Hopelessness (22.7%), and BioSocial Theory (2.3%). Machine learning studies comprised 66 effect sizes.

### Prediction by paradigm

**Prediction of combined theoretically-driven models.** Together, all effect sizes related to theoretically-driven models predicted combined STBs (*w*OR = 1.74; 95% CI, 1.66–1.82; *k* = 264). Suicidal ideation was predicted more accurately (*w*OR = 2.87; 95% CI, 2.66–3.09; k = 87) than attempts (*w*OR = 1.42; 95% CI, 1.34–1.51; k = 99) and death (*w*OR = 1.07; 95% CI, 1.01–1.150, k = 78). Fig 2 depicts all *w*ORs related to all theoretically-driven models by suicidal ideation, suicide attempts, and suicide death. Fig 3 depicts prediction of combined STBs

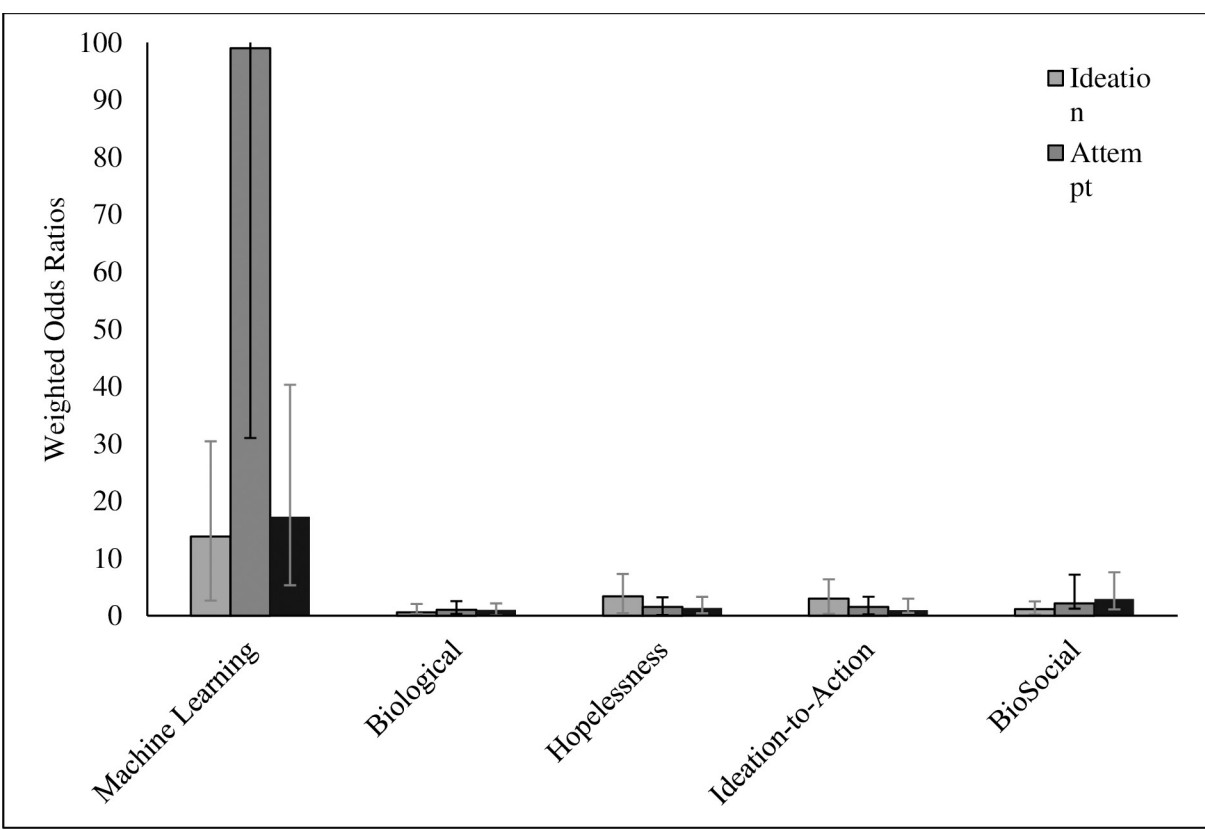

**Fig 2. Prediction of overall, death, attempt, ideation outcomes by theoretical perspective and machine learning models.**

by theory. Heterogeneity was high ($I^2$ = 96.93, Q-value = 8559.78). There was evidence of small sample bias as indicated by Duval and Tweedie's Trim and Fill Test. After trimming 62 effects the true predictive ability of theoretically-driven models was estimated to be $w$OR = 1.05 (95% CI 1.04–1.06). Further, there was evidence of asymmetry as reflected in the Egger's regression intercept test (b = 2.68, 95% CI 2.01–3.35). The funnel plot of small sample bias within the combined theoretically-driven models is depicted in Fig 4.

**Prediction of machine learning models.** Effect sizes related to machine learning indicated that on the whole machine learning approaches provided superior prediction of all suicide-related outcomes ($w$OR = 18.09, 95% CI 15.98–20.47 $k$ = 66). Suicide attempts were predicted with the most accuracy ($w$OR = 99.11, 95% CI 68.91–142.54, $k$ = 27), with prediction of death ($w$OR = 17.29, 95% CI 12.85–23.27, $k$ = 7) and ideation ($w$OR = 13.84, 95% CI 11.95–16.04, $k$ = 33) being similar. Heterogeneity was high ($I^2$ = 98.69%, Q–value = 4973.49). There was evidence of small-sample bias based on the Trim and Fill Method. After trimming two effect sizes the overall predictive ability of machine learning models was estimated to be $w$OR = 17.18 (95% CI 16.91–17.45). Further, as indicated by Egger's regression test (b = 3.44, 95% CI 0.589–6.304), there was evidence of asymmetry. Funnel plot of publication bias of machine learning effect sizes is presented in Fig 5.

Within cross-sectional effect sizes, prediction of combined suicide-related outcomes ($w$OR = 16.45, 95% CI 13.17–20.52, $k$ = 21), suicidal ideation ($w$OR = 16.61, 95% CI 13.57–20.66, $k$ = 20), and suicide attempts ($w$OR = 22.74, 95% CI 9.71–53.23, $k$ = 1) were all statistically significant. Likewise, longitudinal effect sizes predicted combined suicide-related outcomes ($w$OR = 35.82, 95% CI 29–44.14, $k$ = 45), suicidal ideation ($w$OR = 11.28, 95% CI 9.29–

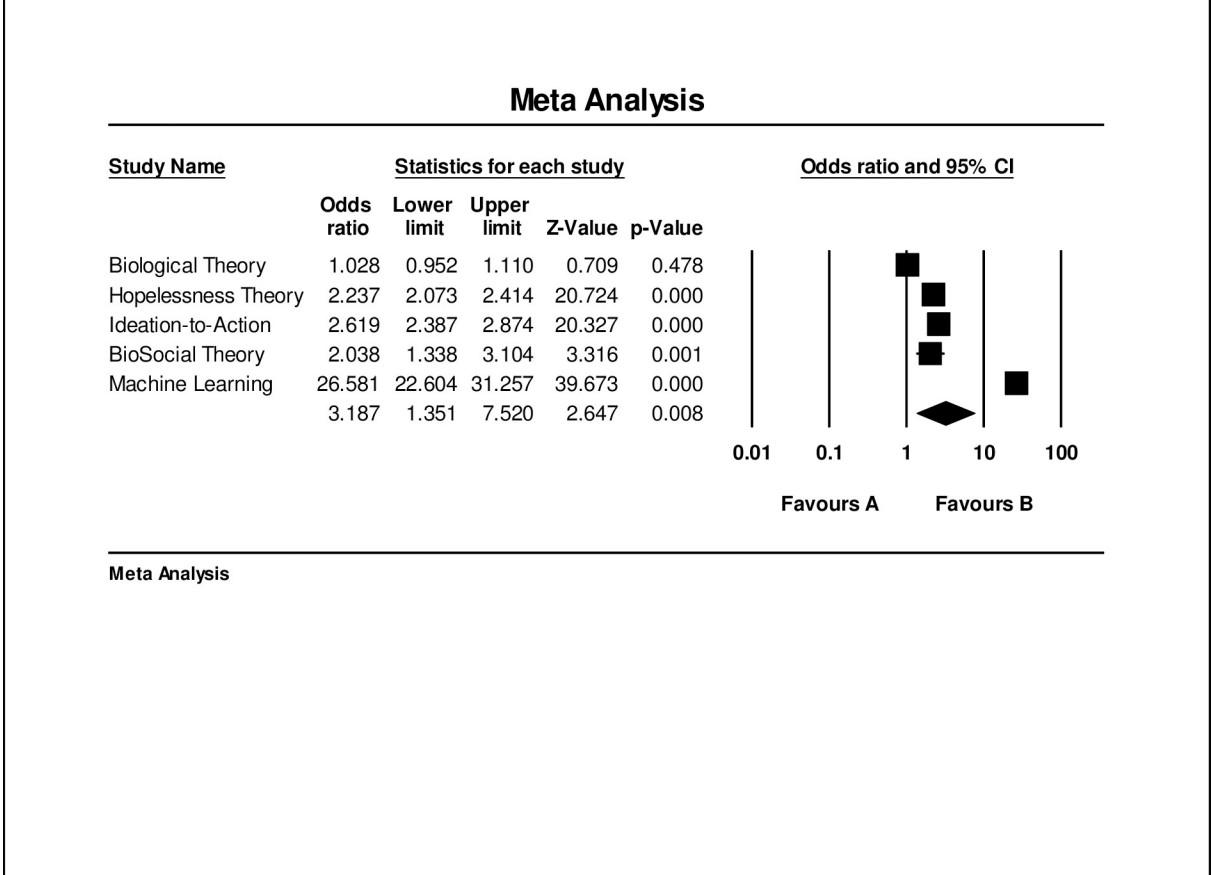

**Meta Analysis**

| Study Name | Statistics for each study | | | | | Odds ratio and 95% CI |
|---|---|---|---|---|---|---|
| | Odds ratio | Lower limit | Upper limit | Z-Value | p-Value | |
| Biological Theory | 1.028 | 0.952 | 1.110 | 0.709 | 0.478 | |
| Hopelessness Theory | 2.237 | 2.073 | 2.414 | 20.724 | 0.000 | |
| Ideation-to-Action | 2.619 | 2.387 | 2.874 | 20.327 | 0.000 | |
| BioSocial Theory | 2.038 | 1.338 | 3.104 | 3.316 | 0.001 | |
| Machine Learning | 26.581 | 22.604 | 31.257 | 39.673 | 0.000 | |
| | 3.187 | 1.351 | 7.520 | 2.647 | 0.008 | |

0.01   0.1   1   10   100

Favours A          Favours B

**Meta Analysis**

**Fig 3. Odds ratios of each paradigm in prediction of combined STB outcomes.**

13.69, $k = 13$), suicide attempts ($w$OR = 121.14, 95% CI 83.46–175.81, $k = 25$), and suicide death ($w$OR = 17.29, 12.85–23.27, $k = 7$) with significance as well. Notably, prediction of longitudinal effect sizes were significantly stronger than cross-sectional effect sizes.

**Comparing the prediction of theoretically-driven models versus machine learning models.** We compared the predictive ability of combined STBs across theories: all theoretically-driven models ($w$OR = 1.74; 95% CI, 1.66–1.82; $k = 264$), Biological ($w$OR = 1.04; 95% CI .97–1.11, $k = 100$), Hopelessness ($w$OR = 1.83, 95% CI 1.71–1.96, $k = 98$), Ideation-to-Action ($w$OR = 2.41, 95% CI = 2.21–2.64, $k = 60$), and BioSocial ($w$OR = 1.32, 95% CI 1.11–1.58, $k = 6$). Notably, Ideation-to-Action predicted combined STBs significantly better than Biological, Hopelessness, and BioSocial Theories. We then compared the theoretically-driven models to machine learning models. Machine learning evidenced superior prediction of combined STB outcomes ($w$OR = 18.09, 95% CI 15.98–20.47 $k = 66$).

Next, we turned our attention to comparing theoretically-driven versus machine learning models as they predict discrete suicide-related outcomes: suicidal ideation, attempts, and death. Findings indicated that machine learning models predicted suicidal ideation ($w$OR = 13.84, 95% CI 11.95–16.04, $k = 33$ versus $w$OR = 2.87; 95% CI, 2.66–1.09; $k = 87$), suicide attempts ($w$OR = 99.11, 95% CI 68.91–142.54, $k = 27$ versus $w$OR = 1.42; 95% CI, 1.34–1.51; $k = 99$), and suicide death ($w$OR = 17.29, 95% CI 12.85–23.27, $k = 7$ versus $w$OR = 1.07; 95% CI, 1.01–1.150, $k = 78$) with more accuracy than combined theoretically driven models.

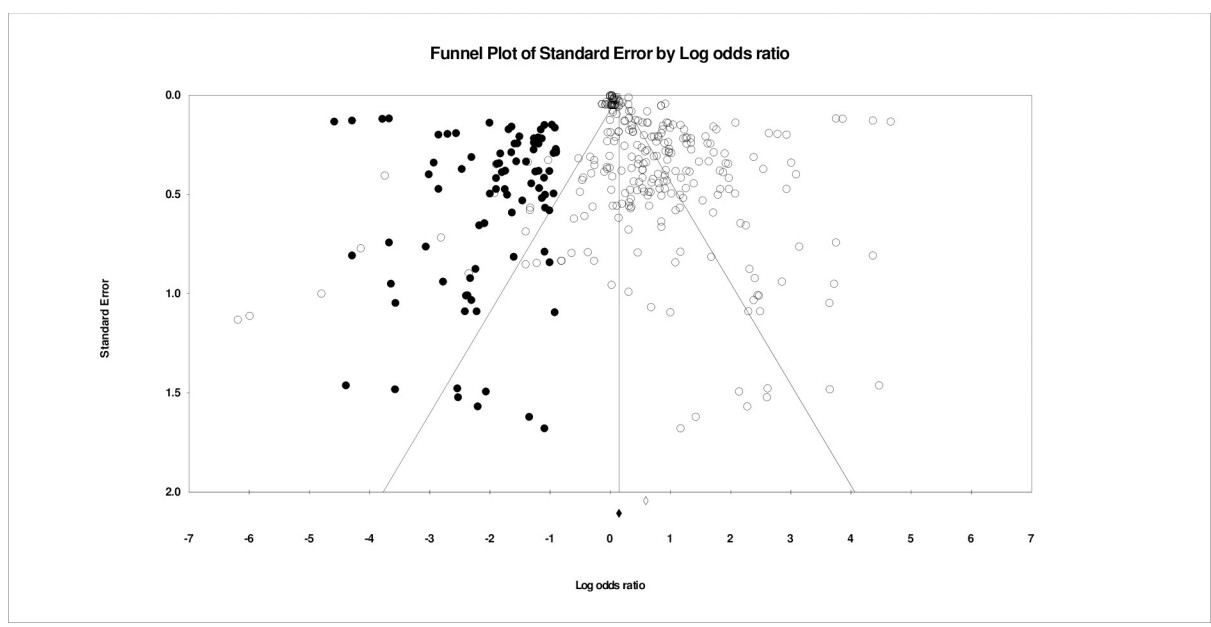

**Fig 4. Publication bias within all effect sizes related to Biological, Hopelessness, Ideation-to-Action, and BioSocial Theories of Suicide.**

**Prediction across theories.** We then investigated each theory on their own to determine if any theory was particularly accurate at predicting combined or discrete suicide-related outcomes. The predictive ability of combined STBs across theories was determined to be: all theoretically-driven models ($w$OR = 1.74; 95% CI, 1.66–1.82; $k$ = 264), Biological ($w$OR = 1.04; 95% CI .97–1.11, $k$ = 100), Hopelessness ($w$OR = 1.83, 95% CI 1.71–1.96, $k$ = 98), Ideation-to-Action ($w$OR = 2.41, 95% CI = 2.21–2.64, $k$ = 60), and BioSocial ($w$OR = 1.32, 95% CI 1.11–1.58, $k$ = 6). Notably, Ideation-to-Action predicted combined STBs significantly better than

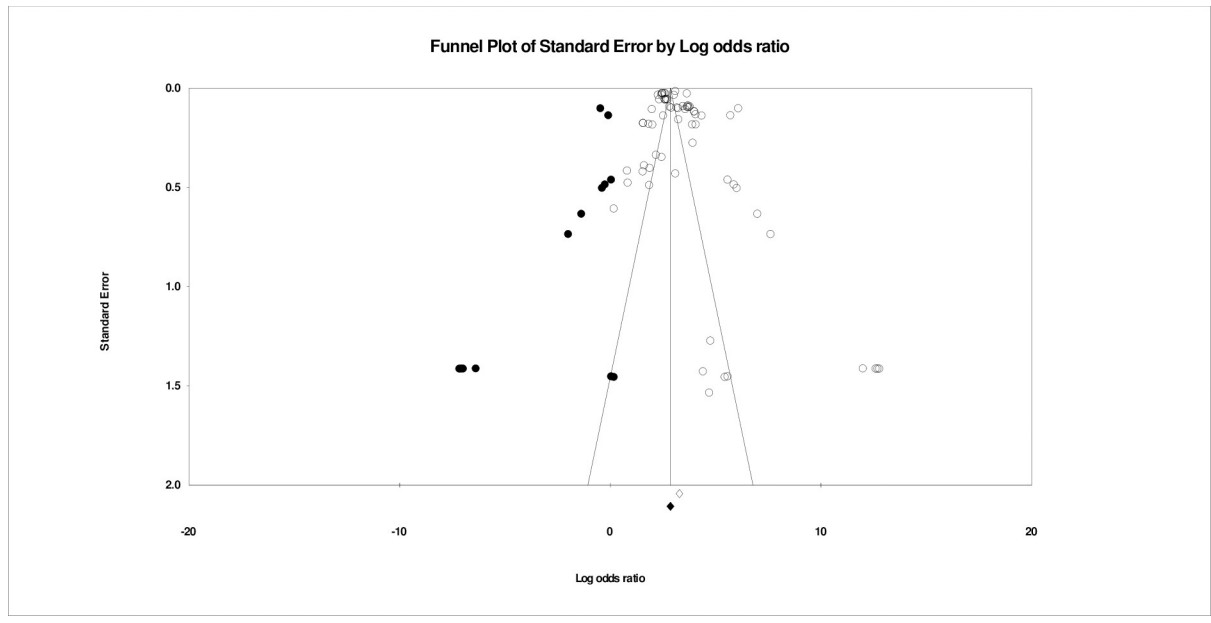

**Fig 5. Publication bias of effect sizes related to machine learning prediction of STBs.**

Biological, Hopelessness, and BioSocial Theories. Also of note, machine learning evidenced superior overall prediction ($w$OR = 18.09, 95% CI 15.98–20.47 $k$ = 66).

**Biological.** Biological effect sizes predicted overall STB outcomes no different than chance ($w$OR = 1.04; 95% CI .97–1.11, $k$ = 100). Within this framework, all discrete suicide-related outcomes were predicted similarly. That is, the Biological theory predicted similarly ideation ($w$OR = .59; 95% CI .25–1.45, $k$ = 4), attempts ($w$OR = 1.04, 95% CI .74–1.47, $k$ = 36), and death (wOR = 1.04, 95% CI .97–1.11, $k$ = 60). Heterogeneity was high ($I^2$ = 81.547 and Q–value = 531.092). Publication bias was indicated in the Trim and Fill plot which reflected that after 12 studies were trimmed the true effect of the Biological Theory was $w$OR = 0.986 (95% CI 0.913–1.06). However, Egger's regression intercept (b = -0.0235, 95% CI -.621–0.578) indicated that the spread of Biological effect sizes was generally symmetrical.

**Hopelessness.** Analyses regarding Hopelessness theory indicated that prediction of combined STB outcomes was superior to Biological Theory ($w$OR = 1.83, 95% CI 1.71–1.96, $k$ = 98). Hopelessness theory predicted suicidal ideation ($w$OR = 3.38, 95% CI 2.91–3.91, $k$ = 35) with significantly more accuracy than attempts ($w$OR = 1.55, 95% CI 1.42–1.67, $k$ = 48) and death ($w$OR = 1.34, 95% CI .93–1.95, $k$ = 15). Heterogeneity was high ($I^2$ = 96.99%; Q–value = 3160.19). Significant publication bias was indicated by Duval and Tweedie's Trim and Fill test. After trimming 40 effect sizes the true estimate of Hopelessness theories was reduced to wOR = 1.05 (95% CI 1.02–1.06). Egger's regression intercept indicated asymmetry with many effect sizes trending towards over inflation of effects (b = 3.257, 95% CI 2.19–4.32).

**Ideation-to-Action.** For overall prediction ($w$OR = 2.41, 95% CI = 2.21–2.64, $k$ = 60), accuracy was superior to Biological and Hopelessness Theories. Ideation-to-Action predicted ideation ($w$OR = 3.05, 95% CI 2.75–3.40, $k$ = 50) significantly better than attempts ($w$OR = 1.52, 95% CI 1.28–1.79, $k$ = 12) or death ($w$OR = .96, 95% CI .46–2.01, $k$ = 1). Between-study heterogeneity was high ($I^2$ = 98.70%, Q–value = 4741.62). There was evidence of publication bias using the Trim and Fill method, which indicated that after trimming two effect sizes the true estimate of Ideation-to-Action was reduced to 1.078 (95% CI 1.07–1.09) when predicting combined suicide-related outcomes. Further, by Egger's regression test (b = 6.352, 95% CI 4.498–8.206) there was significant evidence of asymmetry.

**BioSocial theory.** Effect sizes related to BioSocial Theory were statistically significant predictors of combined STBs ($w$OR = 1.32, 95% CI 1.11–1.58, $k$ = 6). Further, prediction of death was significant ($w$OR = 2.91, 95% CI 1.80–4.69). However, prediction of ideation ($w$OR = 1.127, 95% CI .93–1.37, $k$ = 1) and attempts ($w$OR = 2.179, 95% CI = .95–4.98, k = 2) did not differ from chance. Heterogeneity was high ($I^2$ = 94.48%; Q-value = 90.64). There was no evidence of publication bias using the Trim and Fill method; no effect sizes were trimmed. However, there was evidence of asymmetry as defined by Egger's regression test (b = 3.89, 95% CI 1.103–6.684).

**Prediction trends across years of research.** The first studies predicting suicidal ideation, suicide attempts, or suicide deaths were published in 1976. Meta-regressions based on 1-year intervals indicated a significant change in predictive ability across time for STB outcomes (b = .071, $p < .001$). When we analyzed theoretically-driven models and machine learning models separately, we found that there was a significant improvement in predictive ability across time within the theoretically-driven models (b = .023, $p < .001$) but not within machine learning studies (b = .03, $p = 0.17$).

## Discussion

The aims of the present study were two-fold. We first sought to summarize and compare literature regarding predictive ability of four commonly-studied theories of suicide (*i.e.*, BioSocial,

Biological, Ideation-to-Action, and Hopelessness theories). We then directly compared these theoretically-driven models to machine learning models in their prediction of combined as well as discrete suicide-related outcomes (*i.e.*, suicidal ideation, suicide attempts, and suicide death). Generally, findings indicated that, when combined, theoretically-driven models predicted suicide-related outcomes at similar and suboptimal levels. When analyzed separately, Ideation-to-Action theories emerged as more accurate than BioSocial, Biological, and Hopelessness theories. Finally, machine learning models improved substantially upon theoretically-driven models in the prediction of combined and discrete suicide-related outcomes. Nuances, implications, and considerations of these points are discussed below.

Longstanding theories of suicide have been the status quo of suicidology since at least as early as Durkheim [31]. With regard to more modern study, four popular theories of suicidal behavior have been proposed and have been the center of substantial study: Biological, BioSocial, Ideation-to-Action, and Hopelessness theories. These theories generally propose that a small number of biopsychosocial constructs come together in relatively simple ways to bring about suicidal ideation, attempts, and death. Some recent meta-analyses (*e.g.* [24–29] indicated that biopsychosocial risk factors of STBs are relatively suboptimal predictors. Unfortunately, those meta-analyses were too broad to assess the predictive ability of longstanding theories, and thus the predictive accuracy of longstanding theories of suicide has been left unstudied. Recently, the field of suicidology has welcomed machine learning into the arena of suicide prediction. While machine learning has been the focus of popular meta-analyses (*e.g.* [21–23]), no study has directly compared the predictive ability of longstanding traditional theories of suicide with novel machine learning approaches. The present study addressed these gaps.

Using effect sizes related to all four theories of suicide, the prediction of combined and discrete suicide-related outcomes was similar and suboptimal. That is, when we conducted analyses using effect sizes from all four theories of suicide, analyses indicated that theories of suicide predict suicidal ideation, attempts, and death with similar strength. None of the outcomes were predicted with particularly potent accuracy, and while they were statistically significant, predictions were still very much inaccurate. Further, the strength of the prediction of STB outcomes is suboptimal, falling at near chance-levels. Thus, while theoretically-driven models were statistically significant predictors of STB outcomes, together their predictive ability fell short of highly accurate prediction of any STB outcomes. (Although these findings indicate that leading theories may not have optimal predictive power, it is important to stress that these theories were proposed to uncover the etiology of STBs, with optimal prediction never the goal. This is discussed in more depth below.) In order to compare the relative accuracy of theories, we then analyzed theories separately. The BioSocial, Biological, and Hopelessness theories predicted combined as well as discrete suicide-related outcomes similarly and without optimal accuracy. In other words, none of these theories emerged as particularly accurate predictors of any of the suicide-related outcomes, and again prediction fell near-chance. However, as compared to the Biological, BioSocial, and Hopelessness theories, those that clustered under the Ideation-to-Action framework provided superior prediction of combined STB outcomes. We were surprised to find this significant difference, and upon investigation, found that Ideation-to-Action theories of suicide predicted suicidal ideation with about twice the accuracy of suicide attempts and suicide death. Thus, the increased predictive accuracy of combined suicide-related outcomes on the part of Ideation-to-Action was impressive and likely driven by effect sizes related to the relatively accurate prediction of suicidal ideation. This increased predictive accuracy could indicate that theories subsumed under Ideation-to-Action framework provide substantial prediction *and* understanding of suicidal ideation. That is, theories clustered under the Ideation-to-Action framework (*e.g.*, Interpersonal Theory, 3-Step Theory, and Motivational-Volitional Theory) make strong propositions that constructs such as thwarted

belongingness, perceived burdensomeness, hopelessness about these states, loneliness, entrapment, and loss of connection contribute to the development of suicidal ideation. Our work indicates that the literature generally supports these hypotheses, and it is possible that these constructs may be reliable risk factors, maintaining factors, and treatment targets in patients with suicidal ideation. Alternatively, the superior predictive accuracy of the Ideation-to-Action framework could be reflective of the large number of constructs employed by these theories to predict suicidal ideation. That is, whereas other theories rely on a small set of biopsychosocial constructs to predict suicide-related outcomes, Ideation-to-Action theories are varied and use many different, yet related, independent variables (*e.g.*, connectedness, loneliness, thwarted belongings, entrapment, etc.) to predict suicidal ideation. Thus, it is possible that the relatively accurate prediction of Ideation-to-Action framework is due to the vast and varied constructs used in the predictive models. Further, with regards to all theories of suicide that were studied in our project, there is still substantial progress to be made yet. Even though theories of suicide were statistically significant predictors of STBs, the low base rate of suicide calls into question the clinically significant role that these theories may play in prediction of STBs.

Machine learning models of suicide prediction stood opposite to theoretically-driven models that relied on smaller numbers of constructs and more traditional statistical approaches. The machine learning approach generally used dozens of data points combined in mathematically complex ways to achieve classification of outcomes. Findings from the second aim of our study indicate that, based on extant literature, machine learning models were significantly more accurate than theoretically-driven models across all outcomes: suicidal ideation, suicide attempts, and suicide death. In fact, machine learning prediction of STBs was many times greater than that of traditional, theoretical models. Thus, in the second aim of our project—the direct comparison of predictive accuracy of theoretically-driven versus machine learning models—machine learning models provided significantly improved predictive accuracy of suicidal ideation, suicide attempts, and suicide death as compared to the longstanding theories that have guided the field of suicidology for decades.

Machine learning models are still new to the field of suicide prediction, as compared to longstanding theories that have been studied over the course of decades, and we sought to evaluate aspects of their predictive ability. For example, in our present work we included both cross-sectional ($k = 21$) and longitudinal ($k = 45$) effect sizes of machine learning studies. That is, during initial planning phases of the present project, we were aware of the nascent nature of machine learning within suicidology, initially finding a paucity of longitudinal literature. Thus, in order to increase the number of eligible effect sizes, we opted to include both cross-sectional and longitudinal literature. When we conducted moderator analyses, comparing cross-sectional versus longitudinal effect sizes of machine learning, we found that machine learning models that employed longitudinal, as opposed to cross-sectional, data produced superior classification of discrete and combined suicide-related outcomes. The superior prediction achieved in using longitudinal data could be due, in many cases, to many years' worth of data accruing. For example, it is possible, that patients who experienced suicide-related outcomes had long and extensive self-report, medical, and social media histories indicating distress. Thus, this work provides evidence that past thoughts, emotions, and behaviors could be very much relevant to the accurate prediction of suicide-related outcomes.

The predictive superiority of machine learning represented in our study reflects significant excitement in the field of suicidology. However, while machine learning models were a substantial improvement upon theoretical models of suicide, the state of the literature still left much unclear. For example, relatively few published articles contained data sufficient for analyses ($k = 66$) and when contacted most corresponding authors did not share their data. The few articles that provided sufficient data demonstrated that machine learning models are

significantly more accurate than traditional STB theories, but did not improve in accuracy by publication year. Importantly, the present data indicated significant variability in the accuracy of machine learning models, with highly accurate prediction (*i.e.*, odds ratios greater than 1000) being the exception rather the norm.

Further, it is very much worth noting that machine learning and theoretically-driven models attempt entirely disparate goals. Theoretically-driven models estimate explanatory variance in STBs, whereas most machine learning models optimize equations in the quest for accurate predictive ability. In other words, theories of suicide have been proposed in the effort to *understand* what causes suicidal ideation, attempts, and death. With this goal of understanding the causal mechanisms of suicide-related outcomes comes significant clinical utility. Theories that provide understanding of *how* suicidal ideation, attempts, and death develop, also likely provide clues as to *what* clinicians should target in patients with suicidality. Many theories have the goal of identifying biopsychosocial risk factors that can be identified, targeted, and reduced through evidence-based treatments. However, at great odds to this goal, machine learning models generally do not attempt to understand what causes suicide-related outcomes to occur (with some exceptions, including [31]). Instead, most machine learning models only seek to classify or predict cases as having suicide-related outcome or not. These machine learning models largely do not seek to identify maintaining or causal factors of suicidal ideation, attempts, or death, and in that respect provide less clinical utility than their theoretically-driven counterparts. (See [20] for an in depth discussion of principles, techniques, concepts, and tools the field of psychology can use to balance these two competing goals). Thus, predictive accuracy—the common metric in the present meta-analysis—lends itself to the utility of machine learning models, potentially at the cost of explainability. A notable exception, and perhaps one way to incorporate beneficial aspects of the two approaches (*i.e.*, theoretically-driven models attempt understanding while machine learning models attempt optimal prediction), are machine learning models that use inductive bias (*e.g.*, informative priors in Bayesian models). This data-analytic approach allows for pre-specified knowledge to act as a guide to the data-driven modeling. However, if these pre-determined biases err, *a priori* decisions can be corrected by the machine learning models.

Meta-analyses reflect the state and substance of extant literature, and the extant literature regarding STB prediction from theoretically driven models lends itself to limitations. Present literature largely investigates STB theories from *single* theoretically relevant constructs, not the constellations of constructs that are proposed by theorists. This represents a significant shortcoming in the comparison of theoretical models (often relying on single predictors) to machine learning models (which by their very nature are comprised of large numbers of constructs). Even so, tests of theoretically relevant constructs yielded inaccurate prediction and it is unlikely that combining these inaccurate predictors will result in excellent prediction. To test STB theories, these theories must be directly tested as they were proposed: longitudinally and with combinations of constructs. Machine learning research also has limitations. Few papers employing machine learning models reported sufficient data for analyses. As investigations of STBs using machine learning models grow in number, every subsequent investigation should commit to disclosure of four key values: true positive, true negative, false positive, and false negative identification. These four key values transparently demonstrate relative strengths and weaknesses of model performance. For example, researchers often report model accuracy using area under the curve (*i.e.*, AUC). AUCs take on values between 0 and 100, with more accurate models having AUCs closer to 100. AUCs can be misleading, in that models with high AUCs can, and often do, have relatively poor positive predictive values (*i.e.*, the probability that subjects with a predicted to experience STBs truly have STBs). Researchers and computer scientists will improve comparisons of model accuracy upon the widespread reporting of true positive, true negative, false positive, and false negative.

There was also evidence of significant heterogeneity across the whole of included effect sizes, as well as within theoretically-driven and machine learning models on their own. Although this expansive heterogeneity is not a limitation in its own right, it is indicative of substantial variability in the accuracy of theoretically-driven and machine learning models. This high level of heterogeneity could evidence that while particular theories or machine learning models may be accurate within specific samples, they may not account for the same amount of variance in all cases of suicidal ideation, attempts, and death across the entirety of the population. That is, machine learning models are built to optimally fit specific datasets, thus it is not surprising that one model does not fit all samples. However, high heterogeneity is notable in the case of theories of suicide, which *were* proposed to explain all cases of suicide-related outcomes. As the field moves forward, novel theories of suicide may find increased predictive accuracy by reducing the scope of the populations within which they predict suicide-related outcomes. In essence, this high level of heterogeneity could indicate that there is not a uniform effect across the whole of the human population for each construct or machine learning model as it predicts suicide-related outcomes.

An additional limitation exists particularly with regard to machine learning models. In samples from the present study, machine learning models were trained and tested using samples that were highly similar. Quite often they exhibited strong prediction on unseen samples drawn from the same population as the training sample. However, it is unlikely that machine learning models trained and tested for example, among adolescents, will perform well when applied to a sample of veterans. This limitation–degradation of performance on test data that is substantially different from the training–is a well-known issue in supervised machine learning and the subject of a great deal of research under the general heading of transfer learning [32]. Moreover, in both machine learning and traditional approaches, we found evidence of small sample publication bias. This suggests that smaller samples tend to have much larger effect sizes than larger samples. Therefore, pooled effect size estimates should be interpreted with caution.

Studies were coded based on the quality of their independent variables, such that studies employing objective metrics quantifying independent variables were coded as superior to those employing subjective measures of diagnoses or symptoms. Interestingly, we found a substantial amount of homogeneity within this coding scheme, in that all studies within the BioSocial, Hopelessness, and Ideation-to-Action frameworks relied on self-report, subjective symptom and diagnostic metrics, whereas, studies within the Biological and machine learning approaches both employed quantitative measures of objective independent constructs. This notable difference highlights an inherent divergence in the approaches and propositions of the causes of suicidality, such that it is possible that theories and approaches employing higher quality, objective metrics of independent constructs, may lend themselves particularly well to prediction of STBs as a goal, whereas theories relying on symptoms and diagnostic metrics may provide more actionable constructs for clinicians to target in treatment.

## Future directions

One key to progress in machine learning prediction is community-level work on large shared datasets (*e.g.*, ImageNet for visual object recognition [33], MIMIC III for analysis of electronic health records [34]). A central benefit of shared datasets is that they make possible direct comparisons of competing machine learning models, (e.g. [35]). The use of competing models on common datasets allows variability in the predictive ability of models to be introduced *only* based on models, not on idiosyncratic differences in data. Based on findings from the present literature, it is very likely that the combination of many constructs, as opposed to a single, highly accurate predictors, may be more accurate in the prediction of suicide-related outcomes. It is worth noting that community-level research on prediction of health-related

outcomes has historically been particularly difficult given the sensitive nature of health data, a challenge that has led in mental health to development of datasets involving proxy dependent variables such as *post* hoc human assignments of risk level, self-reports, or behavioral signals (*e.g.*, [35,36] see discussion in [37]). One promising solution to that problem is the development of computation environments that bring researchers to the data rather than placing onus on researchers to acquire and manage data [38].

Another important consideration for future work is the ethics of big data and sensitive data points as related to prediction of suicide, particularly with regard to the idea of deploying such technologies, Ethical issues, guidelines, and best practices are a rapidly growing subject of discussion, particularly in regard to use of non-traditional data sources for machine learning such as social media ([39–41]). Resnik et al. [19] discuss at some length concerns regarding personal autonomy, informed consent, and usage of sensitive data, particularly for methods involving social media sources such as Reddit, Twitter, and Facebook. Some of these concerns do not apply to more traditional approaches in the *study* of suicidology, but they may nonetheless be very much relevant in considering *deployment* of predictive models, whether or not the predictive techniques were developed using traditional theories or machine learning. It is essential that progress on accurate prediction of suicide-related outcomes involve interdisciplinary teams of researchers and include a prominent place for discussions of ethical issues such as privacy, as well as considerations of bias and fairness in automated technologies [38,42,43].

Finally, as computational researchers make progress towards the goal of accurate classification of suicide-related outcomes, working closely with clinical psychologists to provide translational findings within their work needs to be a priority. The goal of accurately classifying suicide-related outcomes is entwined with the goal of providing clinicians with an improved understanding of the etiology, maintaining factors, and treatment targets for suicide-related outcomes. Scientific progress is always an interplay between theory and knowledge, on the one hand, and data, on the other. By facilitating new, data-driven analysis of large datasets, machine learning models have the potential not only to improve prediction, but to help improve insight into the causal mechanisms of suicide-related outcomes and thereby to inform new approaches to evidence-based treatments.

## Conclusions

The present project sought to (1) summarize four leading theories of suicide and compare their accuracy in the prediction of suicidal ideation, attempts and death, and (2) directly compare these theoretically-driven models to machine learning models for the first time. When all theories were combined, prediction of suicide-related outcomes was significant but suboptimal and did not differ between suicidal ideation, attempts, or death. That is, in general, theoretical-models did not provide optimal prediction of any suicide-related outcomes that were studied in this project. Further, when comparing *across* theories, the predictive ability of Biological, BioSocial, and Hopelessness theories of suicide did not vary. However, theories within the Ideation-to-Action framework emerged as nearly twice as accurate as other theories in the longitudinal prediction of combined suicide-related outcomes. This superiority of theories within the Ideation-to-Action framework could be driven by the accuracy in their prediction of suicidal ideation, an outcome which the Ideation-to-Action framework has contributed considerable effort to study.

With regard to the second aim, although theoretically-driven models (especially Ideation-to-Action) were generally statistically significant predictors of suicide-related outcomes, machine learning models provided significantly superior prediction of all outcomes. That is, as compared to theories of suicide, machine learning models provided more accurate prediction of suicidal

ideation, attempts, and death. The field of machine learning as it applies to STB prediction is not a panacea and still has significant limitations. It remains to be seen what aspects of machine learning models contributed to their superior accuracy. Machine learning models did not improve over time, and highly accurate prediction was the exception as opposed to the norm. Nonetheless, the transformational potential of machine learning methods seems clear. The key is to integrate the insights of the traditional approaches with the empirical power of the data-driven techniques. We strongly urge suicidologists to engage in interdisciplinary efforts with researchers in computational science, bioinformatics, and related fields to further the cause of suicide prediction and prevention.

## Supporting information

**S1 Checklist.**
(DOC)

**S1 Table. Model related search terms.**
(DOCX)

**S2 Table. Suicide-related outcome search terms.**
(DOCX)

**S3 Table. Longitudinally relevant search terms.**
(DOCX)

**S1 File.**
(DOCX)

## Author Contributions

**Conceptualization:** Katherine M. Schafer, Grace Kennedy, Austin Gallyer, Philip Resnik.

**Data curation:** Katherine M. Schafer.

**Formal analysis:** Katherine M. Schafer.

**Investigation:** Katherine M. Schafer, Grace Kennedy.

**Methodology:** Katherine M. Schafer, Austin Gallyer, Philip Resnik.

**Project administration:** Katherine M. Schafer.

**Resources:** Katherine M. Schafer.

**Software:** Katherine M. Schafer.

**Visualization:** Katherine M. Schafer.

**Writing – original draft:** Katherine M. Schafer, Grace Kennedy, Austin Gallyer, Philip Resnik.

**Writing – review & editing:** Katherine M. Schafer, Grace Kennedy, Austin Gallyer, Philip Resnik.

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
