## [Decision Letter · Decision Letter 0]

8 Dec 2020

PONE-D-20-21525

A Head-to-Head Comparison of Theory Driven vs Machine Learning Prediction of Suicide: A Meta-Analysis

PLOS ONE

Dear Dr. Schafer,

Thank you for submitting your manuscript to PLOS ONE. After careful consideration, we feel that it has merit but does not fully meet PLOS ONE’s publication criteria as it currently stands. Therefore, we invite you to submit a revised version of the manuscript that addresses the points raised during the review process.

We look forward to receiving your revised manuscript.

Kind regards,

Kyoung-Sae Na, M.D.

Academic Editor

PLOS ONE

Journal Requirements:

2.  Our internal editors have looked over your manuscript and determined that it may be within the scope of our Understanding and Preventing Suicide Call for Papers. This collection of papers is headed by a team of Guest Editors for PLOS ONE: Jo Robinson, Merike Sisask, Kairi Kõlves and Maria Oquendo. With this collection we hope to bring together researchers exploring  suicide and self-harm from societal, psychological, spiritual, clinical, and biological perspectives. Additional information can be found on our announcement page: https://collections.plos.org/s/understanding-suicide. If you would like your manuscript to be considered for this collection, please let us know in your cover letter and we will ensure that your paper is treated as if you were responding to this call. Agreeing to be part of the call-for-papers will not affect the date your manuscript is published. If you would prefer to remove your manuscript from collection consideration, please specify this in the cover letter.

3. Please assess the quality of the studies included in your meta-analysis, as required in item 12 of the PRISMA checklist. Study quality should be presented for each study included in the meta-analysis and the conclusions of your study should take into account the quality of the included studies.

Reviewers' comments:

Reviewer's Responses to Questions

**Comments to the Author**

1. Is the manuscript technically sound, and do the data support the conclusions?

Reviewer #1: Yes

Reviewer #2: Yes

Reviewer #3: Yes

Reviewer #4: Yes

2. Has the statistical analysis been performed appropriately and rigorously? 

Reviewer #1: Yes

Reviewer #2: I Don't Know

Reviewer #3: Yes

Reviewer #4: Yes

3. Have the authors made all data underlying the findings in their manuscript fully available?

Reviewer #1: Yes

Reviewer #2: Yes

Reviewer #3: Yes

Reviewer #4: No

4. Is the manuscript presented in an intelligible fashion and written in standard English?

Reviewer #1: Yes

Reviewer #2: Yes

Reviewer #3: Yes

Reviewer #4: Yes

5. Review Comments to the Author

Reviewer #1: I appreciated the opportunity to review the authors’ meta-analysis comparing theory-driven and ML prediction of suicidal thoughts and behaviors. This fills a gap in the literature that will be useful to suicide researchers, as well as clinicians. Overall, this is a strong manuscript that could benefit from further discussion of ML and, briefly, its limitations prior to the Discussion section, which would be informative in the context of seemingly robust predictive ability and incessantly rising suicide death rates. This is important given that we know accuracy of clinician judgment regarding true suicide risk to be near chance. Specific recommendations may be found below:

INTRODUCTION

1) The authors did well to describe the background of the various theories, as well as the emergence of ML techniques. I believe that it would benefit readers to further describe the process of human-developed ML strategies and the process through which learning/training occurs – this would ultimately provide a basis for a discussion of limitations later in the manuscript.

2) Authors should make clear that ML is not a new technique, though it is slowly working its way into suicidology. It has been used in medical contexts for many years (I recognize this is alluded to in the Discussion).

METHODS – strong section

RESULTS

MACHINE LEARNING

3) Please correct “There was Publication evidence of publication bias…”

4) The Discussion section notes several times that few ML models reported sufficient data for analyses – this may be important to note with numbers in the Results section.

DISCUSSION

5) It may be worthwhile to operationally define ‘suicide prediction’ at the outset, as it is not necessarily true that early theories intended to predict suicide death as much as ‘prevent’ STBs as a whole which, to this day, is also a term with various philosophical definitions.

6) I believe that this could be rephrased in a more accurate and descriptive manner – “This novel approach generally used dozens of features combined in mathematically complicated ways.”

7) I appreciate the discussion of changes, or lack thereof, based on publication date, as well as the disparate goals. Limitations are also generally well-outlined. Life circumstances change over time, such that the moving target of suicide risk is not obviated through ML at its current state, which is focused on a specific target population for each given study using [available] variables chosen by humans. Also, I question whether all theories purport to offer the complex distinctions between the STBs, which is what ML intends to accomplish. While this is touched upon in the Discussion, it is important to ground readers in the Introduction. The Introduction comment about serotonergic activity/alterations is one area of neurobiology that has been under investigation, with interesting findings, but the focus of these studies are arguably different from studies using traditional theories of suicide. Suicide death is a major area of focus within 5-HT system, which is a less common study outcome of focus based on Table 1 of this study.

FUTURE DIRECTIONS

8) It may be worthwhile to note the ethics of big data and sensitive data points (SDOH) – which is necessary for the strongest ML models – if the field is to develop more broad-reaching efforts that are not limited by sample.

OTHER

9) Some references are formatted in different style.

Reviewer #2: Thank you for providing me the opportunity to review this valuable research. To my understanding, the findings of this study are extremely important to the future development of suicide prediction models that could help reduce this tragic phenomenon. However, precisely because I believe that the study is important, I have had many comments and I encourage the authors to conduct major revisions in the current version of the manuscript (see the attached document). Please know, that despite my long list of comments, my unequivocal recommendation to the editor is to accept the manuscript.

Please see the attached review.

Reviewer #3: Summary:

This meta-analysis presents an interesting comparison between 4 traditional theories to predict suicide with the application of maching learning on the same topic. Theories for suicide prevention compared were (i.e. Biological, Hopelessness, Ideation-to-Action, and BioSocial Theories).

It presents an interesting discussion about the results found and the lack of publishing data that allow us to replicate the experiments proposed in some articles.

Figures and Tables: clearly presented and correctly labeled.

Methods: It is recommended to place the start date of the information search in the inclusion criteria.

Regarding the comparison of maching learning models, it would be interesting to know the most frequently used variables.

Study Design and Statistics is clearly explained.

Results:

When starting to write the results it is important to put the amounts of each percentage of the included studies by theory.

Regarding the way of comparing the years, it is recommended that they segment their corresponding amount by traditional theories and those of maching learning.

Discussion:

The comparison of theoretical models (on single predictors) to machine learning models (comprised of large numbers of constructs) shows a great difference in the prediction, on the one hand it can also affect the data used and what the authors report.

Reviewer #4: General

- Some minor grammatical errors

Introduction

- The introduction does a good job stating the rationale for the study and the gap in current research addressed by the meta-analysis.

Methods

- “Literature searches using PubMed, PsycINFO, and Google Scholar were conducted until prior to May 1, 2020” (pp. 7). If all searches were not conducted on the same day, this needs to be explained.

Studies were included if they had “at least one longitudinal analysis predicting suicidal ideation, attempts, or death using any variable related to theoretical constructs” – how did the authors determine whether a variable was “related” to a theoretical construct?

- The data extraction process and analyses are well described.

Results

- There is a discrepancy between the number of studies included in the analyses in the results section and the studies listed in Table 1 (e.g., the authors reported that k = 67 studies were included in the machine learning sub-analyses, but only 18 are listed in Table 1)

Discussion

- The authors included cross-sectional and longitudinal machine learning studies, but only longitudinal traditional model studies. A) More explanation is needed as to why cross-sectional traditional model studies were excluded. B) How many of the machine learning studies were longitudinal vs. cross-sectional? C) It would be helpful to include a sub-group analysis looking at effect sizes for cross-sectional vs. longitudinal machine learning studies.

- The authors report “highly accurate prediction (i.e. odds ratios greater than 1000) being the exception rather the norm” for machine learning studies. Would be helpful to provide a supplemental table listing the ORs for individual studies (in both the machine learning and traditional model subgroups), given that an OR this large could inflate the overall effect size for machine learning.

- “When contacted most corresponding authors did not share their data” (pp. 23). How many authors were contacted for data? How many studies were included after authors provided data? This needs to be indicated in the search strategy section as well as in the PRISMA flowchart.

- Heterogeneity was high in almost all of the analyses. This should be addressed in the discussion.

6. PLOS authors have the option to publish the peer review history of their article (what does this mean?). If published, this will include your full peer review and any attached files.

Reviewer #1: No

Reviewer #2: **Yes: **Yaakov Ophir

Reviewer #3: **Yes: **Gema Castillo-Sánchez

Reviewer #4: No

---

## [Author Response · Author response to Decision Letter 0]

10 Mar 2021

Review on Manuscript PONE-D-20-21525

Response to Reviewers 

Dear Dr. Kyoung-Sae Na, M.D.,

We are pleased to be given the opportunity to submit a revised draft of the manuscript “A Head-to-Head Comparison of Theory Driven vs Machine Learning Prediction of Suicide:

A Meta-Analysis” for publication in PLOS ONE. We appreciate the thoughtful and thorough comments provided by all the reviewers as well as your work as editor. We are grateful for this opportunity to improve the clarity and impact of our work and as such we have incorporated the suggestions made by the reviewers. Those changes are made using the Track Changes feature of Microsoft word. Please see below, for a point-by-point response to the reviewers’ comments and concerns in blue. Again, we greatly appreciate your thoughtful comments as well as those made by all reviewers. Finally, we are happy that this project may fall within the scope of Understanding and Preventing Suicide Call for Papers, and we would like our manuscript to be included in that call.

Thank you for the opportunity to reformat this document. We have updated the document to be consistent with PLOS ONE’s formatting style requirements. 

2. Our internal editors have looked over your manuscript and determined that it may be within the scope of our Understanding and Preventing Suicide Call for Papers. This collection of papers is headed by a team of Guest Editors for PLOS ONE: Jo Robinson, Merike Sisask, Kairi Kõlves and Maria Oquendo. With this collection we hope to bring together researchers exploring suicide and self-harm from societal, psychological, spiritual, clinical, and biological perspectives. Additional information can be found on our announcement page: https://collections.plos.org/s/understanding-suicide. If you would like your manuscript to be considered for this collection, please let us know in your cover letter and we will ensure that your paper is treated as if you were responding to this call. Agreeing to be part of the call-for-papers will not affect the date your manuscript is published. If you would prefer to remove your manuscript from collection consideration, please specify this in the cover letter.

Thank you for this invitation. We would like for our manuscript to be included in the Understanding and Preventing Suicide call for papers. We have included this in our cover letter. 

3. Please assess the quality of the studies included in your meta-analysis, as required in item 12 of the PRISMA checklist. Study quality should be presented for each study included in the meta-analysis and the conclusions of your study should take into account the quality of the included studies.

Thank you for this revision. To meet this aspect of PRISMA guidelines, we have coded study quality based on the type of metric that was used to measure the independent variables. This is consistent with previous meta-analytic literature. Studies that used quantitative, objective metrics of constructs were coded as most optimal (i.e., 1) and studies that were coded using self-report metrics of constructs were coded as less optimal (i.e., 2). These values were added to Table 1 and they have been added into the conclusions of our study found in the Discussion section. 

The project now reads: “Studies were coded based on the quality of their independent variables, such that studies employing objective measures quantifying independent variables were coded as superior to ones employing subjective measures of diagnoses or symptoms. Interestingly, we found a substantial amount of homogeneity within this coding scheme, in that all studies within the BioSocial, Hopelessness, and Ideation-to-Action frameworks relied on self-report, subjective symptom and diagnostic metrics, whereas, studies within the Biological and machine learning approaches both employed quantitative measures of objective independent constructs. This notable difference highlights an inherent divergence in the approaches and propositions of the causes of suicidality, such that it is possible that theories and approaches employing higher quality, objective metrics of independent constructs, may lend themselves particularly well to prediction of STBs as a goal, whereas theories relying on symptoms and diagnostic metrics may provide more actionable constructs for clinicians to target in treatment.”

Thank you for this comment. The corresponding author has updated their account to be linked with ORCID iD.

Thank you for this opportunity to provide captions and to reformat in-text citations within our manuscript. 

Captions within the Supporting Information section now read: “Presented in this Supporting Information are the search terms used for the present meta-analysis. Search terms listed in the “Model Related Search Terms” are related to theoretically-driven or machine learning models. Each search term from the “Model Related Search Terms” list was paired with a search term from the “STB Outcome Search Terms” as well as the “Longitudinally Relevant Search Terms” to achieve strings of search terms.”

Reviewer #1: I appreciated the opportunity to review the authors’ meta-analysis comparing theory-driven and ML prediction of suicidal thoughts and behaviors. This fills a gap in the literature that will be useful to suicide researchers, as well as clinicians. Overall, this is a strong manuscript that could benefit from further discussion of ML and, briefly, its limitations prior to the Discussion section, which would be informative in the context of seemingly robust predictive ability and incessantly rising suicide death rates. This is important given that we know accuracy of clinician judgment regarding true suicide risk to be near chance. Specific recommendations may be found below:

INTRODUCTION

1) The authors did well to describe the background of the various theories, as well as the emergence of ML techniques. I believe that it would benefit readers to further describe the process of human-developed ML strategies and the process through which learning/training occurs – this would ultimately provide a basis for a discussion of limitations later in the manuscript.

Thank you for this comment. We agree that a brief discussion of the development of machine learning will be helpful for the readers. As such we have added the following text to the Introduction section: 

“Generally speaking, machine learning approaches use “training data” to observe which variables (or complex combinations of variables) lead to the most accurate classification of suicide-related outcomes. That is, machines determine complicated algorithms that increase the frequency of “hits” (accurate classification of present or absent suicide-related outcomes) while minimizing false-alarms or missed cases. Notably, machines learn the most optimal algorithms on one set of data, usually referred to as “training data” and then these algorithms are tested on unseen or new data, referred to as “training data.” This is a notable diversion from the theoretically-driven models which do not provide learning and testing on two disparate groups of data. Further, although theoretically-driven models rely on longitudinal studies to test their predictive abilities, machine-learning models can use cross-sectional data in their tests of prediction as they shift from training to testing data. Thus, although theoretically-driven models must use longitudinal data to explore prediction, machine learning models arguably measure prediction as they test their complicated algorithms in new, unseen data in the testing set.”

2) Authors should make clear that ML is not a new technique, though it is slowly working its way into suicidology. It has been used in medical contexts for many years (I recognize this is alluded to in the Discussion).

Thank you for this comment. We have updated our manuscript to reflect that machine learning is not a new technique, but is instead newly applied within the field of suicide prediction.

METHODS – strong section

RESULTS

MACHINE LEARNING

3) Please correct “There was Publication evidence of publication bias…”

Thank you for this correction. The manuscript reflects this change. 

4) The Discussion section notes several times that few ML models reported sufficient data for analyses – this may be important to note with numbers in the Results section.

Thank you for this comment. We have added the following footnote into the Results section. “Of 26 machine learning manuscripts that appeared to meet eligibility criteria but were missing sufficient data, when contacted, only 5 of the corresponding authors replied with sufficient data.”

DISCUSSION

5) It may be worthwhile to operationally define ‘suicide prediction’ at the outset, as it is not necessarily true that early theories intended to predict suicide death as much as ‘prevent’ STBs as a whole which, to this day, is also a term with various philosophical definitions.

Thank you for this comment. This has been added throughout the whole of the document now, including the Introduction and Discussion sections, and includes the following addition: “It is worth noting that machine learning and theoretically-driven models attempt entirely disparate goals. Theoretically-driven models estimate explanatory variance in STBs, whereas most machine learning models optimize constructs in the quest for accurate predictive ability. In other words, theories of suicide have been proposed in the effort to understand what causes suicidal ideation, attempts, and death. However, at great odds to this goal, machine learning models have never attempted to understand what causes suicide-related outcomes to occur. Instead, machine learning models only seek to classify cases as having suicide-related outcome or not.”

6) I believe that this could be rephrased in a more accurate and descriptive manner – “This novel approach generally used dozens of features combined in mathematically complicated ways.”

Thank you for the opportunity to clarify this approach. Throughout the document we have updated the manuscript to reflect a more descriptive and accurate view of machine learning. For example, one of these additions reads: 

“Generally speaking, machine learning approaches use “training data” to observe which variables (or complex combinations of variables) lead to the most accurate classification of suicide-related outcomes. That is, machines determine complicated algorithms that increase the frequency of “hits” (accurate classification of present or absent suicide-related outcomes) while minimizing false-alarms or missed cases.”

7) I appreciate the discussion of changes, or lack thereof, based on publication date, as well as the disparate goals. Limitations are also generally well-outlined. Life circumstances change over time, such that the moving target of suicide risk is not obviated through ML at its current state, which is focused on a specific target population for each given study using [available] variables chosen by humans. Also, I question whether all theories purport to offer the complex distinctions between the STBs, which is what ML intends to accomplish. While this is touched upon in the Discussion, it is important to ground readers in the Introduction. The Introduction comment about serotonergic activity/alterations is one area of neurobiology that has been under investigation, with interesting findings, but the focus of these studies are arguably different from studies using traditional theories of suicide. Suicide death is a major area of focus within 5-HT system, which is a less common study outcome of focus based on Table 1 of this study.

Thank you for the opportunity to weave this into the Introduction. We have updated the manuscript to reflect that most theories generally propose the same, or similar, constructs as causes of suicide-related outcomes. A notable exception to this is the Biological Theory, which represents a departure from this trend, and proposes that constructs related to hormones, neurotransmitters, and specific brain regions may be at play for suicide-related outcomes.

FUTURE DIRECTIONS

8) It may be worthwhile to note the ethics of big data and sensitive data points (SDOH) – which is necessary for the strongest ML models – if the field is to develop more broad-reaching efforts that are not limited by sample.

Thank you for this suggestion. We agree that big data and sensitive data points are a substantial concern for the field of machine learning that may not apply to more traditional theoretically-driven models. This has been discussed at length in another published manuscript by the authors of this paper (see Resnik et al., 2020). We have included a brief description and a citation for readers.

“Another important consideration for future work is the ethics of big data and sensitive data points as related to prediction of suicide, particularly with regard to the idea of deploying such technologies, Ethical issues, guidelines, and best practices are a rapidly growing subject of discussion, particularly in regard to use of non-traditional data sources for machine learning such as social media (Benton, Coppersmith, & Dredze, 2017; Chancellor et al., 2019; Naslund & Aschbrenner, 2019). Resnik et al. (2020) discuss at some length concerns regarding personal autonomy, informed consent, and usage of sensitive data, particularly for methods involving social media sources such as Reddit, Twitter, and Facebook. Some of these concerns do not apply to more traditional approaches in the study of suicidology, but they may nonetheless be very much relevant in considering deployment of predictive models, whether or not the predictive techniques were developed using traditional theories or machine learning. It is essential that progress on accurate prediction of suicide-related outcomes involve interdisciplinary teams of researchers and include a prominent place for discussions of ethical issues such as privacy, as well as considerations of bias and fairness in automated technologies (Abebe et al., 2020; Gebru et al., 2018; Lane & Schur, 2010).”

OTHER

9) Some references are formatted in different style.

Thank you. Those errors have been corrected. All references are formatted in APA formatting.

Reviewer #2: Thank you for providing me the opportunity to review this valuable research. To my understanding, the findings of this study are extremely important to the future development of suicide prediction models that could help reduce this tragic phenomenon. However, precisely because I believe that the study is important, I have had many comments and I encourage the authors to conduct major revisions in the current version of the manuscript. Please know, that despite my long list of comments, my unequivocal recommendation to the editor is to accept the manuscript.

Thank you for your thoughtful comments on our project. We greatly appreciate your thorough read of our work and the opportunity to expand and clarify many of our points. Further, we agree that the requested changes greatly improve our project, its impact, and its overall readability. We have incorporated all the requested edits. 

Abstract

1. To date, a direct, head-to-head comparison…On the one hand, this is a very direct (and therefore informative) description of the research goal. On the other hand, it sounds a bit argumentative/competitive, and I am not sure that this is the most appropriate/tactically-smart narrative. You can consider re-framing it to something like: ML approaches emerged and are said to yield improved predictions but to date, this argument has not been investigated empirically in a systematic manner. The present... a meta-analysis, which allows a direct comparison (without the "head-to-head" phrase).

Thank you for this comment. We agree that the tone of this project is very important and as authors we want to prioritize a healthy discussion of the benefits and limitations to these different approaches. As such, we have reduced the wording throughout that lends itself to a tone of unnecessary competition. The Background of the Abstract now reads as follows: 

“Theoretically driven models of suicidal thoughts and behaviors have long guided research efforts; however, more recently, a machine learning approach to suicide prediction has emerged. Some have suggested that machine learning approaches yield improved predictions but to date, this argument has not been investigated empirically in a systematic manner. The present work, a meta-analysis, presents for the first time a direct comparison of machine learning and theoretically driven approaches.”

2. Conclusions: Among traditional theories, Ideation-to-Action outperformed Biological,

Hopelessness, and BioSocial theories of suicide It is strange to mention a conclusion that is not preceded by relevant findings in the Results section of the abstract. Also, are you sure that Ideation-to-Action outperformed Hopelessness? see a more detailed comment in the Results section.

Again, we thank you for this thoughtful comment. We do agree that, without providing the relevant statistical values in the Results section, the conclusions are a bit unsupported. To support our conclusions, we included the weighted odds ratios for each relevant theory of suicide. Given that the 95% confidence interval for over prediction by Ideation-to-Action does not overlap with those of the other theories, it is our interpretation that the Ideation-to-Action theory does provide superior prediction of suicidal ideation, attempts, and death. Please see the exact wording of the Results and Conclusions sections here.

Results: We analyzed 330 effect sizes. Traditional theories demonstrated weak prediction of ideation (wOR = 2.87; 95% CI, 2.65 – 3.09; k = 87), attempts (wOR =1.43; 95% CI, 1.34 – 1.51; k = 98), and death by suicide (wOR = 1.08; 95% CI, 1.01 – 1.15; k = 78). Generally, Ideation-to-Action (wOR = 2.41, 95% CI = 2.21 – 2.64, k = 60) outperformed Hopelessness (wOR = 1.83, 95% CI 1.71 – 1.96, k = 98), Biological (wOR = 1.04; 95% CI .97 – 1.11, k = 100), and BioSocial (wOR = 1.32, 95% CI 1.11 – 1.58, k = 6) theories of suicide. Machine learning demonstrated superior accuracy of ideation (wOR = 13.84; 95% CI, 11.95 – 16.03; k = 33), attempts (wOR = 99.01; 95% CI, 68.10 – 142.54; k = 27), and death by suicide (wOR = 17.29; 95% CI, 12.85 – 23.27; k = 7). 

Conclusions: Among traditional theories, Ideation-to-Action statistically outperformed Biological, Hopelessness, and BioSocial theories of suicide. However, across all theoretically-driven effect sizes, prediction of suicide-related outcomes was relatively weak. Machine learning models provided substantially superior prediction of suicidal ideation, suicide attempts, and suicide death as compared to theoretically driven models.

Introduction

3. kills nearly one million people every year2 According to the WHO, "Close to 800 000 people die due to suicide every year". See in:https://www.who.int/news-room/factsheets/detail/suicide#:~:text=Close%20to%20800%20000%20people,%2D19%2Dyear%2Dolds.

Thank you for this comment. We agree that the wording of “nearly one million people” is an over-estimation of the actual numerical value. We have amended our phrasing and the document now reads as follows. “Suicide is a leading cause of death worldwide, a global epidemic that kills close to 800,000 people every year (World Health Organization, 2019).”

4. suicide rates have failed to appreciably decline in the United States. I think you should decide whether you want to discuss suicide as a "global epidemic" or as a US phenomenon. If the global framing is used, then, this sentence should be re-phrased or deleted, because the overall picture is mixed. In fact, suicide rates in many many countries have actually fallen! See in: https://ourworldindata.org/suicide#how-have-suicide-rates-changed

Thank you for this point. We very deeply appreciate the disparity in trends of suicide rates among countries around the globe, particularly the rising suicide rates in America as compared to falling rates in other nations. In order to reflect this nuance, the sentence is now changed and reads as follows: “Despite widespread investigation into suicidal thoughts and behaviors and although this is not the case world-wide, suicide rates have failed to appreciably decline in the United States (Centers of Disease Control and Prevention, 2019; WHO, 2019).”

5. Investigations into suicide have traditionally based on one biopsychosocial factor or a small set of factors combined together. Is this the main (and only) difference between the traditional methods and ML methods? If so, it sounds a bit too easy, almost making the research question (what is the superior method) redundant. Of course, any analysis that comprises several predictors will be better than an analysis that has only one predictor. If, indeed, this is the only difference, perhaps you can consider adding a sentence regarding the unique value of using top-down, theory driven predictors in the traditional methods, so that the research problem will be a little bit more complex/interesting.

Thank you for the opportunity to clarify this very nuanced discussion. As the manuscript previously read it did not reflect that there has been a shift in the field. Whereas theories seek to understand the causal mechanisms of suicidal ideation, attempts, and death, machine learning is aimed at optimizing prediction. Fundamentally, the two approaches are attempting different goals. Prior to this revision, these disparate goals were not clarified in the Introduction of the manuscript. We have changed the manuscript and it now reads as follows:

“Fully understanding the causal mechanisms of suicidal ideation, attempts and death has been a long-standing goal for suicide research and practice. Investigations into suicide have traditionally taken the approach of predicting suicidal thoughts and behaviors (STBs) based on one biopsychosocial factor or a small set of factors combined together (e.g. Hopelessness Theory, Beck, 1976; Biological, Oquendo et al., 2014; Ideation-to-Action, Van Orden et al., 2010; Klonsky and May, 2013; O’Connor; 2011; BioSocial Theory of Suicide, Linehan, 1983). However, in recent years, a new approach suicide research and practice has arisen in the field, one aimed at accurate prediction of suicidal ideation, attempts, and death. Optimization of the prediction of suicidal ideation, attempts, and death has been rooted in machine learning. The two differing approaches have yet to be directly compared, head-to-head, using a single metric. The present project addresses this crucial gap.”

We also included this within the Discussion section as well: 

“Further, it is very much worth noting that machine learning and theoretically-driven models attempt entirely disparate goals. Theoretically-driven models estimate explanatory variance in STBs, whereas most machine learning models optimize equations in the quest for accurate predictive ability. In other words, theories of suicide have been proposed in the effort to understand what causes suicidal ideation, attempts, and death. With this goal of understanding the causal mechanisms of suicide-related outcomes comes significant clinical utility. Theories that provide understanding of how suicidal ideation, attempts, and death develop, also likely provide clues as to what clinicians should target in patients with suicidality. Many theories have the goal of identifying biopsychosocial risk factors that can be identified, targeted, and reduced through evidence-based treatments. However, at great odds to this goal, machine learning models generally do not attempt to understand what causes suicide-related outcomes to occur . Instead, most machine learning models only seek to classify or predict cases as having suicide-related outcome or not. These machine learning models largely do not seek to identify maintaining or causal factors of suicidal ideation, attempts, or death, and in that respect provide less clinical utility than their theoretically-driven counterparts . Thus, predictive accuracy—the common metric in the present meta-analysis—lends itself to the utility of machine learning models, potentially at the cost of explainability. A notable exception, and perhaps one way to incorporate beneficial aspects of the two approaches (i.e., theoretically-driven models attempt understanding while machine learning models attempt optimal prediction), are machine learning models that use inductive bias (e.g., informative priors in Bayesian models). This data-analytic approach allows for pre-specified knowledge to act as a guide to the data-driven modeling. However, if these pre-determined biases err, a priori decisions can be corrected by the machine learning models.”

Again, this point is of paramount importance to the document, and the field more generally, and we thank the reviewers for the opportunity to discuss this.

6. The two competing approaches have yet to be directly compared, head-to-head…

See comment 1, above. I think the point is not the lack of head-to-head comparison but a lack of a systematic empirical evidence that ML is indeed a desirable method in this line of research.

Thank you for this comment. We do agree that the wording “head-to-head comparison of machine learning versus theory driven models” may be unnecessarily competitive. Instead, the document now reflects that our project brings forth empirical evidence that machine learning is superior to theory-driven work, when using prediction as a metric. The manuscript now reads:

“With regard to the prediction of suicidal ideation, attempts, and death, there has yet to be a systematic empirical investigation determining if theory-driven versus machine learning models are preferable in this line of research. Thus, the present project addresses this crucial gap.”

7. Theories of Suicide all cluster into (3) the Ideation-to-Action perspective of suicide.

If these theories focus on the shift from ideation to action, then they may not be suitable to the prediction of suicide ideation (only to attempts). See also comment 21.

Thank you for this comment. While their primary focus is to understand the shift from suicidal ideation to suicide attempts, each of the theories proposes risk factors for suicidal ideation. This is addressed more fully in the point below. 

8. However, some recent meta-analyses of risk factors—namely…—have determined that biopsychosocial constructs are suboptimal predictors of STB outcomes. 3 This is a very important statement. However, it does not speak directly with the previous sentence. The four theories have achieved empirical support, however (all kinds of) risk factors are suboptimal. But are these risk factors part of the four theories? The reader does not know. Also, the following sentence says that the four theoretical approaches were not investigated. Thus, leaving the reader a bit perplexed.

Thank you for this comment. We do agree that without a full explanation, the reader is lost. We have updated our manuscript to reflect which sorts of risk factors are associated with the theories. We further clarify the next sentence to indicate that broad/general risk factors are suboptimal. We agree that this is a large point of clarification and we are very grateful to have been given the opportunity to clarify. 

“The Hopelessness, Ideation-to-Action Biological, and BioSocial theories of suicide have each garnered support in the literature. Many studies have indicated that constructs related to Hopelessness theory (i.e., depressive symptoms and hopelessness [e.g., Mars et al., 2019; Soero et al., 2005; Roeder and Cole, 2018]), Biological risk factors (e.g., saturation of serotonin and dopamine [Consoloni et al., 2018; Sher et al., 2006; Yerevanian et al., 2004]), Ideation-to-Action (e.g., entrapment, thwarted belongingness, perceived burdensomeness, loneliness, and lack of connection [e.g., Brent et al., 2005; Chu et al., 2017; Lamis & Lester, 2012]), and BioSocial theory (e.g., dysregulation of negative affect [e.g., Berglund, 1984; Weddig et al., 2012]) all are associated with STBs. However, some recent meta-analyses of broad risk factors—namely internalizing, externalizing, biological, demographics, psychotic features, and non-suicidal self-injury—have determined that biopsychosocial constructs are suboptimal predictors of STB outcomes (Franklin et al., 2017; Huang et al., 2017, 2018; Ribeiro et al., 2016, 2018; Witte et al., 2018).”

8. Franklin et al., 2016 Should be 2017.

Again, thank you for this edit. We have changed this. The manuscript now reflects the appropriate date. 

9. The first aim of the present study…The previous sentence regarding the suboptimal predictors (Franklin et al., 2017) is not really required to present the first aim of the study (comparing the efficacy of the four theoretical approaches). It is more relevant to the second (and from what I understand, the main) goal of the study (proving the importance of ML methods). This also makes the narrative a bit awkward. You may consider breaking the rational into two separate gaps/goals (first, several theories were proposed but with no direct comparison, therefore goal 1. Second, suboptimal predictors can be overcome by using ML - but this is also not proven systematically, therefore goal 2). Alternatively, you can present the two gaps and then present two consecutive goals.

Thank you for this comment. We wholly believe that the readability of this paper is of incredible importance, and we have heeded your recommendations. We restructured the introduction such that we presented the two gaps and then presented the two goals. It is our belief that this restructuring greatly improves the ease of reading for our audience. 

10. Models generally use large datasets…'Models' can refer to traditional statistical analysis as well (e.g., regression model). Please consider adding the term machine learning before the word 'model'. You can also consider using the conventional acronym (ML) throughout the paper.

Thank you for this suggestion. We have reviewed the paper, and for each use of the word “model” we have clarified whether we refer to theoretically versus machine learning models. 

11. (see Belsher et al., 2019) Two following highly relevant meta-analyses are missing in the current version of the paper: 4 Burke, T. A., Ammerman, B. A., & Jacobucci, R. (2019). The use of machine learning in the study of suicidal and non-suicidal self-injurious thoughts and behaviors: A systematic review. Journal of affective disorders, 245, 869-884. Bernert, R. A., Hilberg, A. M., Melia, R., Kim, J. P., Shah, N. H., & Abnousi, F. (2020). Artificial intelligence and suicide prevention: a systematic review of machine learning investigations. International journal of environmental research and public health, 17(16), 5929.

Thank you for these additions. They have been added to the body and Reference sections of the manuscript. These serve to support the readability of the paper, as well as evidence that this a thriving corner of the literature.

12. Thus it is unclear if the accurate novel prediction of machine learning is any improvement upon traditional prediction. In my opinion, this is a more elegant way to present the gap than the 'head-to-head' phrasing that was used in the abstract.

We have continued this wording throughout the paper.

13. The present work represents… of these two competing approaches These are not really competing approaches. Indeed, the current work place the two approaches, one against the other ('head to head), but this does not mean that the approaches are competing, let alone, contradicting.

Thank you for this point. We have updated this. We removed the word “competing” and changed the sentence to the following: “The present work represents a direct comparison of these two approaches: traditional theories and machine learning techniques.”

14. One possibility is that traditional approaches… improve only slightly… An alternative possibility is that theoretical constructs demonstrate strong prediction… consistent with existing theories. Not sure what is the role of these two sentences. First, there is a third option (that traditional approach will not improve the prediction beyond general risk factors). Second, these sentences are not posited as a directional hypothesis (which one of the two options is more reasonable). Third, these options are not backed up with references. So that, in practice, these sentences are not adding any new information beyond the open question (whether the theoretical approaches are valuable or not for suicide prediction).

We have removed the problematic sentences to increase readability.

15. The present work also aims to identify characteristics of highly accurate machine learning models. This is an important goal. However, it comes as a surprise to the reader, as a pleasant 'side effect' of the research. If indeed, you are planning to achieve this goal, please consider building a designated 5 rational and explicitly declare that you are planning to achieve three goals in this study (also in the abstract - currently this goal has no trace in the abstract).

Thank you for this comment. Given that this third goal was not well-developed in the manuscript, we have omitted it all together. We agree that this project is stronger without this third goal.

Method

16. Literature searches… were conducted until prior to May 1, 2020. I have received the manuscript on November 14, 2020. This is quite a long gap. Of course, I do not expect that a new search and a new analysis will be conducted in order to complement these missing, more than six months. However, if you do consider doing that, it would significantly contribute to the comprehensiveness of the study, especially in light of the fact that this field (ML in suicide research) is growing very fast.

Thank you for this comment. We share in your consternation. This manuscript was submitted to PLOS One initially on July 20, 2020 and due to an error with a reviewer, we received communication from PLOS One that it was being reassigned on November 16, 2020. We have included this gap in timing under the limitations section, indicating in particular that this body of literature is rapidly growing. 

17. Finally, we searched the grey literature…Should this expression be explained for readers who are less familiar with meta-analysis conventional practices? (a short description in parenthesis will do the work)

Thank you for this comment. We agree that this expression could be confusing for people less familiar with meta-analyses. The manuscript now reads as follows: Finally, we searched the grey literature by searching for conference postings, reference sections of included papers, and by requesting all papers that were not immediately available via our institutions Interlibrary Loan system.

18. Given the paucity of literature…This is (as will be mentioned in the discussion section) a noteworthy weakness of the study. Perhaps you can elaborate a little about the inherent paradigm of ML methods that distinguish between learning data and test data and on the fact that predictions are made on 'new' unseen data. Indeed, this does not replace the need for longitudinal ML studies, but it distinguishes ML from traditional studies. I believe that it can be asserted that the 'power' of the results from cross-sectional ML designs is different than (superior to) the power of the results from cross-sectional traditional designs. This is because the meaning of the term 'prediction' is somewhat different between the two approaches.

Thank you for this comment. We agree that a brief discussion of training versus testing data, and the role in prediction, will give context to the reader. In the Introduction section we have added the following text to compare this aspect of machine learning models with more theoretically-driven models: 

“Generally speaking, machine learning approaches use “training data” to observe which variables (or complex combinations of variables) lead to the most accurate classification of suicide-related outcomes. That is, machines determine complicated algorithms that increase the frequency of “hits” (accurate classification of present suicide-related outcomes) while minimizing false-alarms or missed cases. Notably, machines learn the most optimal algorithms on one set of data, usually referred to as “training data” and then these algorithms are tested on unseen or new data, referred to as “training data.” Machine learning models can be used cross-sectionally to classify (e.g., identify) participants who endorse suicide-related outcomes at the time of data collection. For example, it is possible for researchers to collect data using multiple psychological self-report measures, physiological measures, and demographic features and construct a machine learning algorithm to classify suicide-related outcomes, all from data collected at the same time point. Alternatively, machine learning models can be used to longitudinally predict who will develop suicide-related outcomes at a point later in time. For example, a machine learning algorithm be trained to use decades of health care or medical records to predict who will develop suicidal ideation at a later point. Thus, machine learning models can be used to classify (cross-sectionally) or predict (longitudinally) cases of suicide-related outcomes. Notably, theoretically-driven models do not provide learning and testing on two disparate groups of data. Further, although theoretically-driven models rely on longitudinal studies to test their predictive abilities, machine-learning models can use cross-sectional data in their tests of prediction as they shift from training to testing data. Thus, although theoretically-driven models must use longitudinal data to explore prediction, machine learning models arguably measure prediction as they test their complicated algorithms in new, unseen data in the testing set.”

Results

19. Ideation-to-Action. For overall prediction (wOR = 2.41…) 6 The overall prediction of ideation to action (wOR) is said to be 2.41 but the specific predictions were all significantly lower (ideation = 2.12, attempts = 1.52, and deaths = 0.96). Does this make sense? This is not the case in the other theoretical approaches (in which the overall figure seems more like a weighted average of the three suicide outcomes). If the overall prediction of ideation to action is not correct, this affects the conclusions of the study and should be fixed throughout the manuscript.

Thank you so much for this opportunity. Upon re-visiting this portion of the results, we realized we did enter an error while transcribing the figures. The incorrect figures related specifically to the prediction of suicidal ideation from Ideation-to-Action theories and have now been amended and the document reads as so:

For overall prediction (wOR = 2.41, 95% CI = 2.21 – 2.64, k = 60), accuracy was superior to Biological and Hopelessness Theories. Ideation-to-Action predicted ideation (wOR = 3.059, 95% CI 2.75-3.40, k = 50) significantly better than attempts (wOR = 1.52, 95% CI 1.28 – 1.79, k = 12) or death (wOR = .96, 95% CI .46 – 2.01, k = 1). Between-study heterogeneity was high (I2 = 98.70%, Q – value = 4741.62) .

20. . Ideation-to-Action predicted ideation (wOR = 2.12…) As mentioned above in comment 7, it is unclear how can a theory that focus on the shift from ideation to action predict ideation?

Thank you for the opportunity to clarify this point. We have amended the framing of this in the Introduction giving more context to these theories. Generally speaking the theories attempt to identify the causes of ideation and then the specific factors that cause ideation to shift toward action. The Introduction now reads as so:

This approach to suicidology seeks to identify the causes of suicidal ideation as well as understand what factors lead to the shift from ideation to action (i.e., suicide attempts and death).

21. Analyses indicated a significant improvement in predictive ability across time when traditional (b = .023, p < .001) and but not within machine learning studies (b =.03, p = 0.17) were analyzed independently. Something is wrong in this sentence.

Thank you for bringing this to our attention. This sentence has been re-worded to be both easy to read and grammatically correct. The manuscript now reads as follows:

“When we analyzed theoretically-driven models and machine learning models separately, we found that there was a significant improvement in predictive ability across time within the theoretically-driven models (b = .023, p < .001) but not within machine learning studies (b =.03, p = 0.17).”

22. Prediction across theories. The paragraph that follows this subtitle does not add new information beyond the results that were presented above.

Thank you for this comment. We agree the placement of this paragraph was odd. We moved it towards to the beginning of the results and then integrated machine learning results, essentially comparing overall prediction of theories to machine learning.

The manuscript now reads as so: We compared the predictive ability of combined STBs across theories: Biological (wOR = 1.04; 95% CI .97 – 1.11, k = 100), Hopelessness (wOR = 1.83, 95% CI 1.71 – 1.96, k = 98), Ideation-to-Action (wOR = 2.41, 95% CI = 2.21 – 2.64, k = 60), and BioSocial (wOR = 1.32, 95% CI 1.11 – 1.58, k = 6). Ideation-to-Action predicted combined STBs significantly better than Biological, Hopelessness, and BioSocial Theories. Machine learning evidenced superior overall prediction (wOR = 18.09, 95% CI 15.98 – 20.47 k = 67).

23. Prediction across suicide outcomes. Not sure whether the paragraph that follows this subtitle adds new information to the results presented above. I do however understand the importance of showing that the CI of the different approaches do not overlap. Perhaps the authors can consider integrating these comparisons within the paragraph that presents the results of the ML approach.

Thank you for this comment. We have reorganized the results to more clearly portray the goals of our study. They flow as so: (1) prediction of all theoretically-driven models, prediction of machine learning models, comparisons of machine learning vs theoretically-driven models, prediction within specific theories, prediction over time.

24. Findings indicated that machine learning models predicted suicidal ideation (…), suicide attempts (…), and suicide death (…). This sentence is not completed. I think the authors meant that ML models predicted suicidal... and suicide death better than traditional approaches.

Thank you for this note. We have revised this sentence and completed as so. The manuscript now reads: Findings indicated that machine learning models predicted suicidal ideation (wOR = 13.84, 95% CI 11.95 – 16.04, k = 33 versus wOR = 2.87; 95% CI, 2.66 – 1.09; k = 87), suicide attempts (wOR = 99.11, 95% CI 68.91 – 142.54, k = 27 versus wOR = 1.42; 95% CI, 1.34 – 1.51; k = 99), and suicide death (wOR = 17.29, 95% CI 12.85 – 23.27, k = 7 versus wOR = 1.07; 95% CI, 1.01 – 1.150, k = 78) with more accuracy than combined theoretically driven models.

25. Positive cases in model testing. 7 The paragraph that follows this subtitle is hard to follow. The goal of the analysis ("investigate the behavior of machine learning models") is obscure and there is no clear rational for this analysis in the introduction. I actually think that this part may be very important but in its current status it only disturbs the reading. Please consider improving this section (and its related parts in the article) or deleting it altogether.

a. First we investigated if the number of positive cases was predictive of accuracy. Findings indicated that it was not (…). I think this last sentence (and the previous one) can be better phrased.

b. We calculated percentage of STB outcomes was calculated as follows… Something is wrong in this sentence.

c. Analyses indicated a statistically significant relation between the proportion of STB cases and accuracy of models (b = .009, p = .0041). This finding should be further explained. What does it mean in laymen language?

Thank you so much. We do agree that this section was problematic and it was removed altogether. 

Discussion

26. Unfortunately, those meta-analyses were too broad…, thus a rush to abandon theories of suicide was premature. Was there really such a 'rush'? If so, a citation is needed.

Thank you for this point. We have changed the wording in this section to remove our previous nod towards a rush to abandon theories. Instead we opted for the following sentence: “Unfortunately, those meta-analyses were too broad to assess the predictive ability of longstanding theories. Recently, the field of suicidology has welcomed machine learning into the arena of suicide prediction. While machine learning has been the focus of popular meta-analyses (e.g. Belsher et al., 2019), no study has directly compared the predictive ability of longstanding traditional theories of suicide with novel machine learning approaches. Thus, the present study addressed these gaps.”

27. The present study first sought to summarize and evaluate extant literature regarding predictive ability of traditional theories of suicide. A similar narrative difficulty appeared in the introduction. This 'first aim' is valid but it is not a natural extension of the previous paragraph. In my opinion, you can open the discussion section by saying that this study aimed to achieve two goals (or three if the last goal is maintained), 1 and 2 and that the findings were... Then you can move to explain why these findings are important (return to the status quo, the rush, the gaps, etc.).

Thank you for this suggestion. We do agree that the structure of the Discussion section posed confusion for the reader. We have reorganized this section to reflect your suggestions, beginning first by restating aims 1 and 2 and following with discussions of each. Further, we have dropped the third aim, as we agree that it was incomplete in our work.

28. This increased predictive accuracy… likely driven by effect sizes related to the Ideation-to-Action framework. 8 Please check this. Something does not add up, not in the numbers presented in the results section and not in the short rational that was supplied for this theoretical framework (see also comments 7 and 21).

Thank you. We have addressed this comment within the Results section. Initially, the values were inaccurate. Further, within the Introduction section we have responded to reviews to more completely and accurately describe Ideation-to-Action theoretical framework. 

29. This stands as a victory for theories clustered under the framework of Ideation-to-Action To my knowledge, this terminology is less acceptable in psychological science (see my related comments on 'competitive' and 'head-to-head').

Thank you again for this comment. Throughout the manuscript we have made substantial efforts to remove the wording that nods to competitive or head to head nature of the work. The project now reflects: “This increased predictive accuracy among theories clustered under the framework of Ideation-to-Action is noteworthy and could indicate that Ideation-to-Action theories provide superior prediction and understanding of the causal influences of suicidal ideation.”

30. That is, the Interpersonal Theory… (all theories in the Ideation-to-Action framework) propose that suicidal ideation precedes serious and/or lethal suicide attempts…Once again, if this is indeed the focus of this approach, how does it contribute to the prediction of ideation itself?

Again, we thank you for the opportunity to expand upon and clarify this point. We have removed the problematic sentence, that in practice added little to the paper. Instead we have opted for the following insertion: “This increased predictive accuracy among theories clustered under the Ideation-to-Action framework is noteworthy and could indicate that Ideation-to-Action theories provide superior prediction and understanding of the causal influences of suicidal ideation. That is, theories clustered under the Ideation-to-Action framework (e.g., Interpersonal Theory, 3-Step Theory, and Motivational-Volitional Theory) make strong propositions that constructs such as thwarted belongingness, perceived burdensomeness, hopelessness about these states, loneliness, entrapment, and loss of connection contribute to the development of suicidal ideation. Our work indicates that the literature supports these hypotheses that seek to estimate the causal contributions of risk factors in the development of suicidal ideation.”

31. . Even though theories… were statistically significant…, the low base rate of suicide calls…This is a good point that, in my opinion, should be further explained, preferably, in the introduction. Elaborating on this point may help readers understand why the small size effects of theory driven approaches are limiting our applicative ability to detect STBs.

Thank you for the opportunity and suggestion to expand upon this point in the introduction. We agree that this is an integral feature as to why theoretically-driven models are less than perfect predictors. We have included the following in the Introduction:

“However, some large, relevant, and recent meta-analyses of broad risk factors—namely internalizing, externalizing, biological, demographics, psychotic features, and non-suicidal self-injury—have determined that biopsychosocial constructs are suboptimal predictors of STB outcomes (Franklin et al., 2017; Huang et al., 2017, 2018; Ribeiro et al., 2016, 2018; Witte et al., 2018). Generally, findings from these meta-analyses indicated that broad risk factors predict suicide-related outcomes with statistical significance, however, effect sizes are relatively low. That is, although broad risk factors are longitudinally related to suicide-related outcomes, due to the low base rates of suicide-related outcomes, these risk factors possess suboptimal clinical utility in the prediction of suicidal ideation, attempts, and death.”

32. For example, few published articles contained data sufficient for analyses and when contacted most corresponding authors did not share their data. Do you mean insufficient?

We thank you for this point. We have changed the wording to improve clarity for the audience. The project now reads: “For example, relatively few published articles contained data sufficient for analyses and when contacted most corresponding authors did not share their data.”

33. It was expected that the percentage of cases (i.e. STB positive participants…) As mentioned above, this is an interesting point. However, it is not developed enough. Why was this the expectation (in the introduction - perhaps provide examples from other fields, perhaps discuss the base-rate problem, etc.) and how come this expectation was not realized (in the discussion). Alternatively, perhaps omit this part (and the entire third goal of the study) altogether.

We appreciate the opportunity to improve our paper. As such we have opted to omit this section and the third goal entirely. Thank you for your feedback. 

34. Further exploratory moderation analyses… were not conducted as too few articles provided sufficient data. 9 This sounds like an 'apology' for not fulfilling an action that was not fully introduced and rationalized to begin with (in the introduction). This is another example of the incompleteness of the 'third goal' of the study.

Thank you for your insight into the incomplete nature of the third goal of the study. The third goal has been removed to improve readability and impact of our work.

35. Thus, it largely remains unclear why some machine learning models predict extremely well and others do not. This is actually a good start of developing the rationale behind the third goal of the study.

Thank you for this comment. Given the underdeveloped nature of this “third goal” we removed it entirely from our project. Thus this rationale, albeit helpful for the third goal, was removed. 

36. Thus, it was unclear if, only by including longitudinal prediction, machine learning accuracy would decrease. I expect that ML models will only benefit from longitudinal designs, but I understand the proposed limitation.

Thank you for this comment. We agree that this is a question still to be answered in the field and we are excited to watch for these findings. 

37. Thus, predictive accuracy—the common metric in the present meta-analysis—lends itself to the utility of machine learning models, potentially at the cost of explainability. This is an interesting point. Please consider elaborating on it because for readers who are not familiar with ML strategies, this distinction is not intuitive - how come prediction and explainability come, one at the expanse of the other? After all, the more the variable explains the criterion, the more it is expected to contribute to its prediction.

We greatly appreciate the opportunity to expand upon this point. We do agree wholly that this is an important point, and as such we have added the following work: “It is worth noting that machine learning and theoretically-driven models attempt entirely disparate goals. Theoretically-driven models estimate explanatory variance in STBs, whereas most machine learning models optimize constructs in the quest for accurate predictive ability. In other words, theories of suicide have been proposed in the effort to understand what causes suicidal ideation, attempts, and death. However, at great odds to this goal, machine learning models have never attempted to understand what causes suicide-related outcomes to occur. Instead, machine learning models only seek to classify cases as having suicide-related outcome or not.”

38. A notable exception, and perhaps a middle ground to the conflict between the two approaches, A 'middle ground' goes back to the 'war' between the two approaches. If you think that combining theory and ML worth further investigation, perhaps phrase it as such (extracting the benefits from each approach). However, please remember, that in its current form, the 'benefits' of theory-driven approaches are not provided/convincing in the introduction section.

Thank you for the opportunity to clarify this point throughout the manuscript. We fully believe that theoretically-driven models of suicide have merit, and as such we have included this throughout the Introduction Specifically, we have included that theories “seek to understand the causal mechanism underpinning the etiology of suicidal ideation, attempts, and death.” This goal is an important component of suicidology research and practice and is not attempted using purely prediction machine learning models

We have also updated the competitive nature of the wording in the Discussion section. It now reads as follows: “A notable exception, and perhaps one way to incorporate aspects of the two approaches, are machine learning models that use inductive bias (e.g. informative priors in Bayesian models). This data-analytic approach allows for pre-specified knowledge to act as a guide to the data-driven modeling. However, if these pre-determined biases err, a priori decisions can be corrected by the machine learning models.”

39. However, it is unlikely that machine learning models trained and tested for example, among adolescents, will perform with high accuracy prediction among a sample of veterans. Is this a unique limitation of ML? Is it not a problem for traditional approaches as well?

Thank you for the opportunity to expand upon this point. This limitation actually is unique to ML models and does not exist within theoretically-driven models. Theories generally suggest that the same set of risk factors leads to suicidal ideation, attempts, and death in very sample. Thus, the theoretically-driven models do not vary between samples of the population. We have included these brief discussion points and the manuscript now reads as follows:

“This limitation is unique to machine learning models and generally does not apply to theoretically-driven models. That is, the theories included in the present analyses are notable for their testable and falsifiable propositions regarding the underlying causal mechanisms of all cases of suicide. The Biological, BioSocial, Hopelessness, and Ideation-to-Action theories are proposed to be equally accurate in all populations, including diverse populations such as LGBTQ+, veteran and active duty military, veterinarians, and adolescents. Theories do not specify risk factors that would be missing or different in any particular sample. Thus, the theoretically-driven models that are accurate among veterans would be expected to be accurate in adolescents, for example.”

40. This suggests that smaller samples tend to have much larger effect sizes than smaller samples. 10 Should the second "smaller" be "larger"? (the word "smaller" appears twice in this sentence).

Thank you for this comment. You are correct in this edit, and our manuscript now reflects this change: “Moreover, in both machine learning and traditional approaches, we found evidence of small sample publication bias. This suggests that smaller samples tend to have much larger effect sizes than larger samples. Therefore, pooled effect size estimates should be interpreted with caution.”

41. In summary, prediction of STB outcomes… In my opinion, you do not need to provide a summary paragraph here. You can continue (from the limitation section) straight to the future directions section, especially since you have a conclusion section by the end of the article.

Thank you for this suggestion. We agree that in order to enhance the reading experience for our audience, this summary paragraph is unnecessary and as such we have removed this section.

Conclusions

42. The conclusions regarding the first goal of the study (i.e., the effectiveness of the four theoretical approaches) are missing from this section. The third goal (which, as mentioned above, is not fully described in the manuscript) is also missing from this conclusion.

Again, we thank you for the opportunity to improve our piece. We have included a summary/description of the findings related to aim 1. Notably, we have chosen to omit aim 3. “Generally, the predictive accuracy of suicide-related outcomes did not vary between the Biological, BioSocial, Hopelessness, and Ideation-to-Action theories of suicide. Further, when employing theoretically-driven models, prediction did not differ between suicidal ideation, attempts, or death. However, one notable exception emerged. Ideation-to-Action predicted suicidal ideation at roughly twice the accuracy as other theories, and this likely reflects accurate hypotheses in these theories related to the causal development of suicidal ideation.”

43. The key is to integrate the insights of the traditional approaches with the empirical power of the data-driven techniques. This is the first time this recommendation is mentioned (the one that proposed the 'middle ground' from above does not imply the benefits of such integration). As mentioned above, if you think that this is an important recommendation, please consider elaborating on it earlier in the discussion section.

Thank you for this opportunity. Through the revisions process we sought to elaborate on the point that theory driven and ML models both present respective strengths. Theories attempt to understand causal mechanisms while ML models attempt accurate classification. This theme is now echoed within the Introduction section and the Discussion section. Within the Introduction, we discuss the strengths of theories as attempting to understand causal mechanisms, whereas ML models seek to optimally predict. Further, the respective benefits of theory driven and ML models are discussed in the Discussion section. For example, the Discussion now reads “A notable exception, and perhaps one way to incorporate beneficial aspects of the two approaches (i.e., theoretically-driven models attempt understanding while machine learning models attempt optimal prediction), are machine learning models that use inductive bias (e.g. informative priors in Bayesian models).” It is our understanding that these efforts now give context to the attempt to integrate the attractive qualities of both the ML and theory driven models.

Reviewer #3: Summary:

This meta-analysis presents an interesting comparison between 4 traditional theories to predict suicide with the application of maching learning on the same topic. Theories for suicide prevention compared were (i.e. Biological, Hopelessness, Ideation-to-Action, and BioSocial Theories). It presents an interesting discussion about the results found and the lack of publishing data that allow us to replicate the experiments proposed in some articles.

Figures and Tables: clearly presented and correctly labeled.

Thank you for these comments.

Methods: It is recommended to place the start date of the information search in the inclusion criteria.

Thank you for this suggestion. The project now reflects these values and reads: “Literature searches using PubMed, PsycINFO, and Google Scholar were conducted on a single day and included any studies published between January 1, 1900 to May 1, 2020.”

Regarding the comparison of machine learning models, it would be interesting to know the most frequently used variables.

We do agree that this aspect of machine learning models would be interesting for the readers and the authors alike. However, most articles employing machine learning techniques did not list or disclose the constructs they used and thus we are unable to report this information. 

Study Design and Statistics is clearly explained.

Thank you for this comment.

Results:

When starting to write the results it is important to put the amounts of each percentage of the included studies by theory.

Included in our project, the most common theories were Biological theories (37.8%) followed by Ideation-to-Action (37.1%), Hopelessness (22.7%), and BioSocial Theory (2.3%).

Regarding the way of comparing the years, it is recommended that they segment their corresponding amount by traditional theories and those of machine learning.

Thank you for this suggestion. The project now reflects this additional analysis. “Analyses indicated a significant improvement in predictive ability across time when traditional (b = .023, p < .001) and but not within machine learning studies (b =.03, p = 0.17) were analyzed independently.”

Discussion:

The comparison of theoretical models (on single predictors) to machine learning models (comprised of large numbers of constructs) shows a great difference in the prediction, on the one hand it can also affect the data used and what the authors report.

Thank you for this comment. Throughout the manuscript, we have expanded upon the discrepant roles that machine learning and theoretically-driven models hold in the literature. Whereas machine learning models seek optimal prediction, theories seek understanding of causality. This is reflected throughout the Introduction and Discussion sections.

Reviewer #4: General

- Some minor grammatical errors

Thank you for this comment. We have read the document repeatedly and thoroughly, removing grammatical errors throughout. 

Introduction

- The introduction does a good job stating the rationale for the study and the gap in current research addressed by the meta-analysis.

Thank you for this comment. 

Methods

- “Literature searches using PubMed, PsycINFO, and Google Scholar were conducted until prior to May 1, 2020” (pp. 7). If all searches were not conducted on the same day, this needs to be explained. 

Thank you for this comment. Searches were all conducted on the same day and included studies that were published until May 1, 2020. The manuscript now reflects this.

Studies were included if they had “at least one longitudinal analysis predicting suicidal ideation, attempts, or death using any variable related to theoretical constructs” – how did the authors determine whether a variable was “related” to a theoretical construct?

This has now been discussed more fully in the Introduction section. We have included a discussion of theories and their related constructs. The manuscript now reads: “Many studies have indicated that constructs related to Hopelessness theory (i.e., depressive symptoms and hopelessness [e.g., Mars et al., 2019; Soero et al., 2005; Roeder and Cole, 2018]), Biological risk factors (e.g., saturation of serotonin and dopamine [Consoloni et al., 2018; Sher et al., 2006; Yerevanian et al., 2004]), Ideation-to-Action (e.g., entrapment, thwarted belongingness, perceived burdensomeness, loneliness, acquired capability, and lack of connection [e.g., Brent et al., 2005; Chu et al., 2017; Lamis & Lester, 2012]), and BioSocial theory (e.g., emotion dysregulation, dysregulation of negative affect [e.g., Berglund, 1984; Weddig et al., 2012]) are cross-sectionally and longitudinally associated with STBs.”

- The data extraction process and analyses are well described.

Thank you for this comment. 

Results

- There is a discrepancy between the number of studies included in the analyses in the results section and the studies listed in Table 1 (e.g., the authors reported that k = 67 studies were included in the machine learning sub-analyses, but only 18 are listed in Table 1)

Thank you for this comment. We have amended this, and Table 1 accurately portrays the studies.

Discussion

A) The authors included cross-sectional and longitudinal machine learning studies, but only longitudinal traditional model studies. More explanation is needed as to why cross-sectional traditional model studies were excluded. 

Thank you for this comment. We appreciate the opportunity to expand upon this point. We have included the following text in the manuscript: “That is, to date, no manuscripts have compared the meta-analytic accuracy of longitudinal investigations of theoretically-relevant constructs grouped together as theories propose. It is of paramount importance that meta-analyses include only longitudinal investigations, as theories were proposed in efforts to understand causal mechanisms that lead to the development of suicide-related outcomes. As cross-sectional studies do not estimate predictive accuracy, they contribute relatively little understanding of the causal role of theoretically-relevant risk factors for suicidal ideation, attempts, and death.” 

B) How many of the machine learning studies were longitudinal vs. cross-sectional? It would be helpful to include a sub-group analysis looking at effect sizes for cross-sectional vs. longitudinal machine learning studies. The authors report “highly accurate prediction (i.e. odds ratios greater than 1000) being the exception rather the norm” for machine learning studies. Would be helpful to provide a supplemental table listing the ORs for individual studies (in both the machine learning and traditional model subgroups), given that an OR this large could inflate the overall effect size for machine learning.

Thank you for this comment. We have updated the project to include analyses of machine learning models that compare accuracy of cross-sectional versus longitudinal approaches. These analyses and their related interpretations are included in the Results and Discussion sections. 

Within the Results section: “Within cross-sectional effect sizes, prediction of combined suicide-related outcomes (wOR = 16.45, 95% CI 13.17 – 20.52, k = 21), suicidal ideation (wOR = 16.61, 95% CI 13.57 – 20.66, k = 20), and suicide attempts (wOR = 22.74, 95% CI 9.71 – 53.23, k = 1) were all statistically significant. Likewise, longitudinal effect sizes predicted combined suicide-related outcomes (wOR = 35.82, 95% CI 29 – 44.14, k = 45), suicidal ideation (wOR = 11.28, 95% CI 9.29 – 13.69, k = 13), suicide attempts (wOR = 121.14, 95% CI 83.46 – 175.81, k = 25), and suicide death (wOR = 17.29, 12.85 – 23.27, k = 7) with significance as well. Notably, prediction of longitudinal effect sizes were significantly stronger than cross-sectional effect sizes.”

Within the Discussion “Machine learning is still new to the field of suicide prediction, as compared to longstanding theories that have accrued study over the course of decades. As we sought to evaluate aspects of this relatively novel approach and its application to suicidology, we included both cross-sectional (k = 21) and longitudinal (k = 45) effect sizes. That is, during initial planning phases of the present project, authors expressed concerns about the nascent nature of machine learning within suicidology, initially finding a paucity of longitudinal literature. Thus, in order to increase the number of viable effect sizes, we opted to include both cross-sectional and longitudinal literature. We found that machine learning models that employed longitudinal data produced superior classification of discrete and combined suicide related outcomes as compared to those that employed cross-sectional data. The superior prediction achieved in using longitudinal data could be due to, in many cases, decades worth of data accruing. It is possible, that for patients who experienced suicide-related outcomes had long and extensive self-report, medical, and social media indicating distress over the course of time. Thus, this work provides evidence that past thoughts, emotions, and behaviors are relevant to the prediction of suicide-related outcomes.” 

D) “When contacted most corresponding authors did not share their data” (pp. 23). How many authors were contacted for data? How many studies were included after authors provided data? This needs to be indicated in the search strategy section as well as in the PRISMA flowchart.

Thank you for this suggestion. This has information has now been included in the Results section and in the PRISMA diagram. Within the Results section, we have added the footnote: “Of an additional 26 machine learning manuscripts that appeared to meet eligibility criteria but were missing sufficient data, when contacted, only 5 of the corresponding authors replied with sufficient data.” Within the PRISMA diagram these values are reflected in the articles that did not meet eligibility based on insufficient data.

- Heterogeneity was high in almost all of the analyses. This should be addressed in the discussion.

Thank you for this comment. We agree that the significant amount of heterogeneity needed discussion. We have included the following text:

“There was also evidence of substantial heterogeneity across the whole of included effect sizes as well as within theoretically-driven and machine learning models on their own. Although this expansive heterogeneity is not a limitation in its own right, it is indicative of substantial variability in the accuracy of theoretically-driven and machine learning models. This could evidence that while particular theories or machine learning models may be highly accurate within specific samples, they may not accurately account for all cases of suicidal ideation, attempts, and death across the entirety of the population. Therefore, we were unable to identify particular moderators that allowed for substantially accurate prediction of suicide-related outcomes, either within theories or within machine learning models.”

---

## [Decision Letter · Decision Letter 1]

26 Mar 2021

A Direct Comparison of Theory-Driven and Machine Learning Prediction of Suicide: A Meta-Analysis

PONE-D-20-21525R1

Dear Dr. Schafer,

We’re pleased to inform you that your manuscript has been judged scientifically suitable for publication and will be formally accepted for publication once it meets all outstanding technical requirements.

Kind regards,

Kyoung-Sae Na, M.D.

Academic Editor

PLOS ONE

Additional Editor Comments (optional):

Reviewers' comments:

Reviewer's Responses to Questions

**Comments to the Author**

1. If the authors have adequately addressed your comments raised in a previous round of review and you feel that this manuscript is now acceptable for publication, you may indicate that here to bypass the “Comments to the Author” section, enter your conflict of interest statement in the “Confidential to Editor” section, and submit your "Accept" recommendation.

Reviewer #1: All comments have been addressed

Reviewer #2: All comments have been addressed

2. Is the manuscript technically sound, and do the data support the conclusions?

Reviewer #1: Yes

Reviewer #2: Yes

3. Has the statistical analysis been performed appropriately and rigorously? 

Reviewer #1: Yes

Reviewer #2: I Don't Know

4. Have the authors made all data underlying the findings in their manuscript fully available?

Reviewer #1: Yes

Reviewer #2: Yes

5. Is the manuscript presented in an intelligible fashion and written in standard English?

Reviewer #1: Yes

Reviewer #2: Yes

6. Review Comments to the Author

Reviewer #1: I appreciate the time taken by the authors to integrate reviewer comments. I believe that my comments have been sufficiently incorporated, along with the comments of others, resulting in a more robust and technically sound manuscript.

Reviewer #2: To the authors,

I thank you again for providing me the opportunity to review your work. The revised manuscript is extremely readable, and I have really enjoyed reading it. From my perspective, the article is "ready to go". In its current form, I think that the article fills an important empirical brick, which was missing from the literature that advocates the integration of Machine Learning (ML) in suicide research.

On that note, I propose that, in the final version, you would acknowledge previous works that raised concerns regarding the superiority and reliability of ML in suicide/mental health research. In a way, your work addresses these concerns (but please remember to maintain the respected tone that is now so well-integrated in the revised manuscript).

Attached are two references for example.

Siddaway, A. P., Quinlivan, L., Kapur, N., O'Connor, R. C., & de Beurs, D. (2020). Cautions, concerns, and future directions for using machine learning in relation to mental health problems and clinical and forensic risks: A brief comment on “Model complexity improves the prediction of nonsuicidal self-injury”(Fox et al., 2019).

Jacobucci, R., Littlefield, A. K., Millner, A. J., Kleiman, E. M., & Steinley, D. (2021). Evidence of Inflated Prediction Performance: A Commentary on Machine Learning and Suicide Research. Clinical Psychological Science, 2167702620954216.

Altogether, I believe your work is solid and highly important and I truly hope to see it published soon.

Best regards,

Yaakov

7. PLOS authors have the option to publish the peer review history of their article (what does this mean?). If published, this will include your full peer review and any attached files.

Reviewer #1: No

Reviewer #2: **Yes: **Yaakov Ophir

---

## [Editor Report · Acceptance letter]

30 Mar 2021

PONE-D-20-21525R1 

A Direct Comparison of Theory-Driven and Machine Learning Prediction of Suicide: A Meta-Analysis 

Dear Dr. Schafer:

I'm pleased to inform you that your manuscript has been deemed suitable for publication in PLOS ONE. Congratulations! Your manuscript is now with our production department. 

Kind regards, 

on behalf of

Dr. Kyoung-Sae Na 

Academic Editor

PLOS ONE